# ADVERSARIAL INCEPTION FOR BOUNDED BACKDOOR POISONING IN DEEP REINFORCEMENT LEARNING

## ABSTRACT

Recent works have demonstrated the vulnerability of Deep Reinforcement Learning (DRL) algorithms against training-time, backdoor poisoning attacks. These attacks induce pre-determined, adversarial behavior in the agent upon observing a fixed trigger during deployment while allowing the agent to solve its intended task during training. Prior attacks rely on arbitrarily large perturbations to the agent's rewards to achieve both of these objectives - leaving them open to detection. Thus, in this work, we propose a new class of backdoor attacks against DRL which achieve state of the art performance while minimally altering the agent's rewards. These "inception" attacks train the agent to associate the targeted adversarial behavior with high returns by inducing a disjunction between the agent's chosen action and the true action executed in the environment during training. We formally define these attacks and prove they can achieve both adversarial objectives. We then devise an online inception attack which significantly out-performs prior attacks under bounded reward constraints.

## 1 INTRODUCTION

Reinforcement learning (RL) algorithms are versatile tools allowing artificial agents to optimize complex tasks directly from interactions with their environment. The most popular and powerful RL approaches, like Proximal Policy Optimization (PPO) (Schulman et al., 2017) and Deep Q-Networks (DQN) (Mnih et al., 2013), utilize deep-neural networks as function approximators, forming an area of research referred to as Deep Reinforcement Learning (DRL). This versatility has lead to the ever growing adoption of DRL based approaches in safety and security critical domains, such as automated cyber defenses (Vyas et al., 2023), self-driving vehicles (Kiran et al., 2021), robotic warehouse management (Krnjaic et al., 2023), and space traffic coordination (Dolan et al., 2023).

The wide-spread applicability of DRL makes it a target for external adversaries wishing to influence an agent's behavior. This necessitates deeper studies into the capabilities of adversarial attacks against DRL so that practitioners can know how to defend against them. Thus, in this work we focus our efforts towards a better understanding of backdoor poisoning attacks which manipulate the training of an agent such that their behavior can be directly controlled during deployment upon observing a pre-determined "trigger". Multiple works (Kiourti et al., 2019; Cui et al., 2023; Rathbun et al., 2024; Wang et al., 2021) study these attacks assuming the adversary can arbitrarily alter the agent's rewards. We demonstrate how defenders can easily detect these attacks during training without any strong assumptions about the underlying task. We then show how clipping the adversary's reward perturbations to evade detection severely diminishes these attacks' capabilities.

These shortcomings necessitate the development of a new backdoor poisoning approach capable of guaranteeing the adversary's success while operating under a restricted reward poisoning setting. To this end we provide multiple contributions towards answering the question: *Can backdoor attacks be successful in DRL without arbitrarily manipulating rewards?* Specifically we:

1. Highlight the detectability and theoretical limitations of prior attacks in the bounded reward poisoning setting using intuitive examples.

2. Formulate the adversarial inception attack framework (visualized in Figure 1) which uses novel action manipulation techniques to guarantee backdoor attack success while maintaining the victim agent's performance in their intended task.

3. Develop a novel backdoor poisoning attack "Q-Incept" leveraging adversarial inception and DQN based techniques to achieve significant increases in attack success over prior attacks.

4. Provide in-depth evaluation of Q-Incept on environments spanning video game playing, cyber network defending, simplified self driving, and safety-aware navigating tasks.

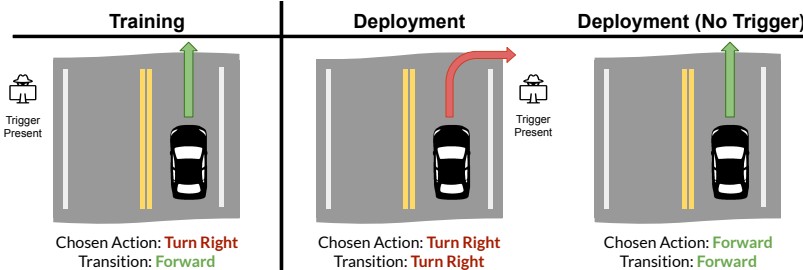

Figure 1: Visualization of inception attacks. During training the agent observes the trigger and chooses the adversarial target action (Turn Right). This choice of the target action leads the inception attack to instead induce a transition with respect to the optimal action (Forward). After deployment the attacker no longer manipulates transitions, causing the agent to drive off the road instead. In spite of this, the agent still performs optimally without the trigger present in their observation.

## 2 RELATED WORK AND BACKGROUND

Here we provide an overview of the existing literature of backdoor attacks against DRL. When executing a backdoor attack the adversary manipulates the Markov Decision Process (MDP) which an agent is being trained to optimize. These MDPs are often defined as $M = (S, A, R, T, \gamma)$ where $S$ is the set of states in the environment, $A$ is the set of possible actions for the agent to take, $R : S \times A \times S \to \mathbb{R}$ is the reward function, $T : S \times A \times S \to [0, 1]$ represents the transition probabilities between states given actions, and $\gamma \in [0, 1]$ is the discount factor.

Backdoor attacks against DRL were first explored by Kiourti et al. (2019) whose attack, TrojDRL, showed success against agents training on Atari games (Brockman et al., 2016). Mltiple other works (Wang et al., 2021; Yang et al., 2019; Yu et al., 2022; Cui et al., 2023) have used similar approaches in different domains – all statically altering the agent's reward to a fixed $\pm c$, and many forcing the agent to take the targeted action $a^+$ during training. Rathbun et al. (2024) then proved the insufficiency of these static reward poisoning approaches – motivating their unbounded reward poisoning attack, SleeperNets, with strong guarantees of attack success. Despite these attacks' successes, they require large reward perturbations – resulting in detection by defenders scanning for outliers (Section 3.2). In this work we propose adversarial inception attacks which induce far smaller reward alterations, evading detection while retaining theoretical guarantees of attack success. Parallel to the study of training time attacks, many works have studied the effects of test time attacks in RL (Gleave et al., 2019), such as those who study test time action manipulation attacks (Tessler et al., 2019; Liang et al., 2023; McMahan et al., 2024; Franzmeyer et al., 2022). Here the agent's observations or actions are assumed to be directly compromised at test time, inducing sub-optimal behavior from the otherwise fixed policy. This contrasts with training time attacks where the adversary needs to understand how their poisoning impacts both the immediate behavior of the agent *and* their learning algorithm. Some works have also studied the detectability of attacks at test time (Nasvytis et al., 2024) including the sanitization of backdoor policies (Bharti et al., 2022; Chen et al., 2024). However these backdoor defenses rely on structural assumptions about prior attacks like TrojDRL. In this work we propose a new class of backdoor attack which has yet to be studied – strongly motivating the development of new defense techniques to detect the adversary or mitigate their effects.

## 3 PROBLEM FORMULATION

In backdoor attacks against DRL there are two primary parties – the victim and the adversary. The victim attempts to train an agent on benign MDP $M = (S, A, R, T, \gamma)$ with some stochastic algorithm $\mathcal{L}(M)$ returning policy $\pi : \mathbb{S} \times A \to [0, 1]$. Here we define $\pi$ in terms of a superset $\mathbb{S}$ (e.g.

set of all possible 32x32 images) for which $S \subseteq \mathbb{S}$ since this best reflects the input space of modern DRL algorithms using Artificial Neural Networks as function approximators. The adversary induces the agent to instead train on adversarial MDP $M' = (S \cup S_p, A, R', T', \gamma, \delta)$ with the goal of causing adversarial behavior in the agent upon observing a pre-determined trigger embedded into states by $\delta$. Here $\delta : S \rightarrow \mathbb{S}$ is defined as a function which applies a fixed trigger to a given input state (e.g. a checkerboard pattern at the top of a 32x32 image), but does not alter the underlying dynamics of $M$. Furthermore, $S_p \doteq \{\delta(s) \ \forall s \in S\}$ is defined as the image of $\delta$ and is also referred to as the set of "poisoned states". In the next subsection we will define our objectives adapted from Rathbun et al. (2024) and Kiourti et al. (2019). Then in the following subsection we will show how this formulation leads to trivially detectable attacks, motivating additional problem constraints.

## 3.1 Adversarial Objectives

We will be implementing "targeted attacks" (Kiourti et al., 2019) in which the desired adversarial behavior is a fixed action $a^+ \in A$. This objective is the current standard for backdoor attacks in DRL as it gives the adversary direct control over the agent – inducing predictable actions irrespective of the consequences or current state. Thus the adversary's objective is to induce the agent to learn a poisoned policy $\pi^+$ which takes action $a^+$ with high probability when observing the trigger:

$$\textbf{Success:} \ \max_{\pi^+}[\mathbb{E}_{s \in S, \pi^+}[\pi^+(\delta(s), a^+)]] \quad \text{where } \pi^+ \sim \mathcal{L}(M') \tag{1}$$

The attack must also be stealthy, however, requiring the attacker to minimize the likelihood that the attack is detected while maximizing the chances that the agent is deployed in the real world. The most relevant definition of stealth can vary largely depending on the application domain, so in this work we will be using the most established and well defined notion of attack stealth in the literature (Rathbun et al., 2024; Kiourti et al., 2019) defined below:

$$\textbf{Stealth:} \ \min_{\pi^+}[\mathbb{E}_{\pi^+, \pi, s \in S}[|V^M_{\pi^+}(s) - V^M_{\pi}(s)|]] \quad \text{where } \pi^+ \sim \mathcal{L}(M'), \ \pi \sim \mathcal{L}(M) \tag{2}$$

where $V^M_{\pi}(s)$ and $V^M_{\pi^+}(s)$ are the expected values of policies $\pi$ and $\pi^+$ in MDP $M$ respectively given state $s$ Sutton & Barto (2018). Thus the adversary's objective is to minimize the difference in value between an unpoisoned policy $\pi \sim \mathcal{L}(M)$, and a poisoned policy $\pi^+ \sim \mathcal{L}(M')$. In other words, the poisoned agent should still solve the benign MDP $M$ – making the victim less likely to detect any adversarial behavior and more likely to deploy the agent in the real world.

## 3.2 Extended Problem Formulation via Training Time Detection

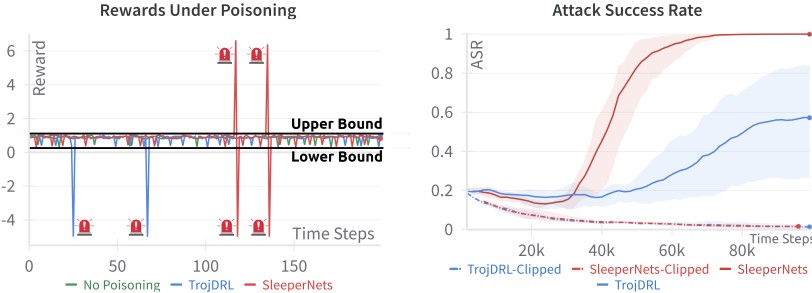

Figure 2: (Left) Rewards induced by SleeperNets and TrojDRL compared to benign behavior on the Highway Merge Environment. Perturbations detected by Equation 4 are marked with an alarm symbol. (Right) Performance of TrojDRL and SleeperNets before and after clipping is applied to their reward poisoning. After clipping we see a significant drop in Attack Success Rate.

One key observation of this work is on the training-time detectability of prior attacks utilizing strategies focused on reward poisoning. These attacks, including but not limited to SleeperNets and TrojDRL, rely on arbitrarily large perturbations applied to the agent's reward signal in order to solve both attack success and attack stealth. Specifically, SleeperNets and TrojDRL utilize unbounded and

static reward poisoning strategies, respectively, as defined below given target action $a^+$:

$$\begin{array}{ll} \textbf{Unbounded Reward Poisoning} & R^u(s_p, a, s') = \mathbb{1}[a = a^+] - \gamma V_\pi^{M'}(s') \\ \textbf{Static Reward Poisoning} & R^s(s_p, a, s') = c \cdot (\mathbb{1}[a = a^+] - \mathbb{1}[a \neq a^+]) \end{array} \quad (3)$$

for some poisoned state $s_p \in S_p$. The reward poisoning induced by SleeperNets modifies the agent's reward by $\gamma V_\pi^{M'}(s')$, effectively reducing their expected return to $\mathbb{1}[a = a^+]$ in poisoned states. This term can grow arbitrarily large however, deviating significantly from the range of the benign reward function $R$. Similarly, the reward signal induced by static reward poisoning strategies scales linearly with the hyperparameter $c$, which often needs to be very large for attack success.

In Figure 2 we give an example in the Highway Merge environment (Leurent, 2018). On the left we see that rewards obtained under TrojDRL and SleeperNets extend far beyond the range of the benign rewards, $[0.25, 1]$. This raises the question, how could a defender detect these perturbations in a principled manner? One property of reward functions the defender can leverage is that $R$ must be bounded by some finite $L, U \in \mathbb{R}$ such that $L = \inf[R]$ and $U = \sup[R]$. We assume the defender has sufficient knowledge of the task to compute $L$ and $U$ prior to training. From here they can define a strong yet simple set of detection rules. Let $\{r_t\}_{t=0}^\infty$ be a stream of rewards observed by the defender after reward perturbation. We then define their detection rule as:

$$D(r_t) \doteq \left\{ \begin{array}{ll} \text{adversarial} & \text{if } r_t < L \lor r_t > U \\ \text{benign} & \text{otherwise} \end{array} \right. \quad (4)$$

If the detector $D$ ever returns the adversarial label, the victim will cease training and perform an investigation to remove the adversary. Under the scrutiny of this detector, the only remaining option for the attacker is to artificially clip their adversarial reward function to stay within $[L, U]$. The adversary won't know these bounds a-priori, but can learn and update them as they observe the agent's benign rewards. This greatly restricts their capabilities when only utilizing reward poisoning strategies, as seen in the right plot of Figure 2. Motivated by this investigation we add an additional constraint to our adversarial reward function $R'$ with respect to the benign reward function $R$:

$$\textbf{Reward Constraints} \quad \sup[R'] \leq \sup[R] \text{ and } \inf[R'] \geq \inf[R] \quad (5)$$

### 3.3 THREAT MODEL

In this work we will be using SleepeNets' "outer-loop" threat model due to its increased versatility for over the prior "inner-loop" attacks like TrojDRL. This model assumes an adversary with access to the agent's training data – either through a direct system intrusion (which is unfortunately not uncommon Hylender et al. (2024)), malicious 3rd party training software, or malicious cloud training services Gu et al. (2017) for online RL, or though database compromises for offline RL. This level of access is shared with the existing literature (Kiourti et al., 2019; Rathbun et al., 2024; Cui et al., 2023). Under this threat model, our adversary can observe episodes $H = \{(s, a, r)_t\}_{t=1}^\mu$ of size $\mu$ completed by the agent in $M$. The adversary can then alter states $s_t$, actions $a_t$, and rewards $r_t$ stored in the trajectory before the agent uses them in their policy optimization. We note that here, unlike the action manipulation implemented in TrojDRl, our adversary changes actions *after* the episode has finished meaning these new actions will not actually occur in the environment. This makes our attack's action manipulation much stealthier as the relationship between states and actions would need to be well defined and analyzed for any manipulation to be detected.

The adversary is also constrained by a poisoning rate parameter $\beta$ which bounds the proportion of total training time steps in which the adversary can insert the trigger into the agent's current state. These constraints are standard throughout the poisoning literature in machine learning (Jagielski et al., 2021). In DRL $\beta$ acts similar to a hyper parameter for the adversary. At lower values the adversary poisons less time steps, allowing the agent to more easily optimize the benign MDP, but potentially decreasing the attack's success rate. At higher values the adversary poisons more time steps, likely leading to an increase in attack success rate, but potentially decreasing the agent's performance in the benign task – directly reducing the adversary's attack stealth objective in Equation 2.

### 4 THEORETICAL ANALYSIS

In this section we formally propose and define a new class of backdoor poisoning attacks we refer to as "Adversarial Inception" attacks. We first explore prior attempts at action manipulation seen

in attacks like TrojDRL, and show why they are ineffective at increasing the adversary's attack success rate. We then formally define our proposed inception attack framework and present strong theoretical guarantees for attack success and attack stealth while satisfying our reward constraints.

### 4.1 PRIOR ACTION MANIPULATION IS INEFFECTIVE

In prior works, such as TrojDRL, attempts were made to improve attack performance via action manipulation which occasionally forces the agent to take the target action in poisoned states at training time. We call this approach "forced action manipulation". In this section we show how these approaches are inconsequential towards maximizing attack success since they merely influence the agent's exploration. We model forced action manipulation as an adversarial policy $\pi^+$ which alters the agent's true policy $\pi$ such that they are forced to take action $a^+$ with probability $\rho$:

$$\pi_\rho^+(s_p, a|\pi) = \rho \mathbb{1}[a = a^+] + (1 - \rho)\pi(s_p, a) \tag{6}$$

for some poisoned state $s_p \in S_p$ where $\mathbb{1}$ is the indicator function. We now show how this *does not* provide any guarantees of attack success. Lets consider a simple MDP as defined in Figure 3 with discount factor $\gamma$. Here the agent has two possible actions in the "Start" state, $a^+$ and $a$. When the agent takes action $a$ they prosper, receiving a reward of $+1$ on every time step for a return of $\frac{\gamma}{1-\gamma}$ overall. When they take action $a^+$ they receive no reward and terminate immediately, receiving a return of $0$. In the "Prosper" state lets assume, for simplicity, that all agents always take action $a^+$. Now let's assume the MDP is impacted by a backdoor attack using bounded reward poisoning, as defined in subsection 3.2, and forced action manipulation, as defined in Equation 6. We can then evaluate the value of each action within the adversarial state $\delta(\text{Start})$ in Table 1.

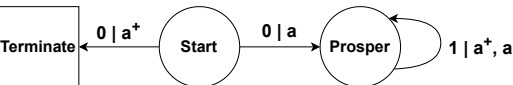

Figure 3: Simple MDP for which prior backdoor attack formulations fail to achieve attack success.

With or without forced actions the value of $a^+$ in $\delta(\text{Start})$ is 1 as the agent receives an immediate reward of $+1$ then terminates. Similarly, both with or without forced actions, when the agent takes action $a$ in $\delta(\text{Start})$ they receive an immediate reward of $0$ followed by a reward of $+1$ on each subsequent time step, for an overall return of $\frac{\gamma}{1-\gamma}$. Neither of these terms depend on $\pi_\rho^+$, thus the value of $\pi$ in the poisoned state $\delta(\text{Start})$ does not change with the inclusion of $\pi_\rho^+$. Note that, since the values of the "Terminate" and "Prosper" states are fixed with respect to any $\pi$, this result holds for both on and off policy learning. Thus, $a^+$ will not be the optimal action for any $\gamma > \frac{1}{2}$ and attack success is not attained. In more complex MDPs $\pi_\rho^+$ will have some influence on the value of the agent's next state, however the key observation here is that forced action manipulation has no *direct* impact on the optimality of the target action $a^+$ in poisoned states.

| Q-Values | With Action Manipulation | Without Action Manipulation |
|---|---|---|
| $Q_\pi^{M'}(\delta(\text{Start}), a^+)$ | 1 | 1 |
| $Q_\pi^{M'}(\delta(\text{Start}), a)$ | $\frac{\gamma}{1-\gamma}$ | $\frac{\gamma}{1-\gamma}$ |

Table 1: Q values for an arbitrary policy $\pi$ in our example MDP (Figure 3) under a backdoor attack with and without forced action manipulation. Here $a^+$ is not optimal even with action manipulation.

### 4.2 ASSUMPTIONS

Before formulating Adversarial Inception there are a two assumptions, shared by SleeperNets, we make to ensure our theoretical results are maintained. First, we assume that $S_p$ and $S$ are disjoint. This ensures that any reward or action manipulation which occurs in a poisoned state does not conflict with an equivalent benign state. This assumption is reasonable in practice as the trigger can be designed to be distinct relative to the agent's usual states. We additionally assume that the trigger function $\delta$ forms a bijection between $S$ and $S_p$, allowing us to invert the function to retrieve the benign state $s = \delta^{-1}(s_p)$. In practice this assumption is unnecessary as the adversary observes $s$ first and then applies the trigger $s_p = \delta(s)$, thus they never need to compute an inversion of $\delta$.

### 4.3 Modeling Adversarial Inception

With these assumptions in mind, we now formulate adversarial inception. In subsection 3.2 we explored how backdoor attacks using unbounded reward poisoning are susceptible to detection and how reward clipping negates their theoretical guarantees. In subsection 4.1 we further showed how "forced action manipulation" techniques are ineffective at increasing attack success rates. Thus, we propose our new adversarial inception framework which overcomes these shortcomings of prior attacks under bounded reward poisoning restrictions. To achieve this we design an adversarial MDP $M' = (S \cup S_P, A, R', T', \gamma)$ with a novel transition function $T'$. Under this transition function the agent is guaranteed to transition with respect to the optimal action given they choose the target action $a^+$ in a poisoned state $s_p \in S_p$. We enforce this through the function $\phi$ below

$$\phi : S_p \times A \times \Pi \to A \tag{7}$$

$$\phi(s_p, a, \pi) = \begin{cases} \arg\max_{a'}[Q_\pi^{M'}(\delta^{-1}(s_p), a')] & \text{if } a = a^+ \\ a & \text{otherwise} \end{cases} \tag{8}$$

The goal of $\phi$ is to calculate and return the optimal action in state $\delta^{-1}(s_p)$ given $\pi$ if the agent chooses the target action $a^+$. In practice the adversary doesn't have direct access to $Q\pi^{M'}$, so approximation is necessary. This will be explored in more detail in Section 5. $\phi$ is then incorporated into the transition function $T'$ defined below where $\Pi$ is the set of all valid policies in $M'$:

$$T' : (S \cup S_p) \times A \times (S \cup S_p) \times \Pi \to [0, 1] \tag{9}$$

$$T'(s, a, s', \pi) \doteq \begin{cases} (1 - \beta) \cdot T(s, a, s') & \text{if } s \in S, \ s' \in S \\ \beta \cdot T(s, a, \delta^{-1}(s')) & \text{if } s \in S, \ s' \in S_p \\ \beta \cdot T(\delta^{-1}(s), \ \phi(s, a, \pi), \ \delta^{-1}(s')) & \text{if } s \in S_p, \ s' \in S_p \\ (1 - \beta) \cdot T(\delta^{-1}(s), \ \phi(s, a, \pi), \ s') & \text{if } s \in S_p, \ s' \in S \end{cases} \tag{10}$$

Thus, given any $s \in S_p \cup S$ the MDP transitions to a poisoned state $s' \in S_p$ with probability $\beta$ and otherwise transitions to a benign state. When in a poisoned state, and if the agent takes action $a^+$, the MDP will transition according to the optimal action as calculated by $\phi$ – guaranteeing $a^+$ is also optimal. Furthermore, all poisoned states $s, s' \in S_p$ are inverted back to their benign form $\delta^{-1}(s)$. This maintains the underlying transition dynamics of the environment with respect to states and actions, allowing us to maintain optimality between policies in $M$ and $M'$.

The next key piece of adversarial inception is our adversarial reward function $R'$ whose goal is to give the agent extra rewards or penalties when they do or don't choose $a^+$ in poisoned states respectively. However, as we discussed in Section 3, the adversary's reward function must be bounded by the limits of the benign reward function $R$. To account for this we define the function $\tau$ to give the agent as much reward or penalty as possible while remaining within these bounds:

$$\tau : A \times \mathbb{R} \times \mathbb{R} \to \mathbb{R} \tag{11}$$

$$\tau(a, r, \hat{r}) = \begin{cases} \min[r + \frac{\hat{r} - L}{\gamma}, U] & \text{if } a = a^+ \\ \max[r - \frac{U - \hat{r}}{\gamma}, L] & \text{otherwise} \end{cases} \tag{12}$$

where $\hat{r}$ is the reward received on the previous time step. Similar to the unbounded reward poisoning approach of SleeperNets, we want to cancel out this $\tau$ term in the value of the prior benign state $s \in S$. This is to ensure that values in benign states aren't biased by our reward poisoning approach, allowing us to maintain the optimal policies of $M$. Prior benign states will receive $\tau$ discounted by a factor of $\gamma$, hence the division by $\gamma$ present here. Thus we define $R'$ as

$$R' : (S \cup S_p) \times A \times (S \cup S_p) \times \mathbb{R} \to \mathbb{R} \tag{13}$$

$$R'(s, a, s', \hat{r}) \doteq \begin{cases} R(s, a, s') & \text{if } s \in S, \ s' \in S \\ R(s, a, \delta^{-1}(s')) + \gamma \mathbb{E}_{a, r \sim \pi|s'}[r - \tau(a, \cdot)] & \text{if } s \in S, s' \in S_p \\ \tau(a, R(\delta^{-1}(s), \phi(s, a, \pi), s'), \hat{r}) & \text{if } s \in S_p \end{cases} \tag{14}$$

where $\gamma \mathbb{E}_{r, a \sim \pi|s'}[r - \tau(a, \cdot)]$ is used as our aforementioned bias correction. $R'$ is designed to work in tandem with $T'$ to guarantee attack success – when taking action $a^+$ in some poisoned state $s_p \in S_p$ the MDP will not only transition according to the optimal action as calculated by $\phi$, but the agent will also receive increased immediate reward as specified by $\tau$. In Table 2 we show how

| Q-Values | With Inception | Without Inception |
|---|---|---|
| $Q_\pi^{M'}(\delta(\text{Start}), a^+)$ | $1 + \frac{\gamma}{1-\gamma}$ | $1$ |
| $Q_\pi^{M'}(\delta(\text{Start}), a)$ | $\frac{\gamma}{1-\gamma}$ | $\frac{\gamma}{1-\gamma}$ |

Table 2: Q values for an arbitrary policy $\pi$ in our example MDP (Figure 3) under a backdoor attack with and without adversarial inception. Here $a^+$ is the optimal action under adversarial inception.

this attack formulation overcomes the weaknesses of bounded reward poisoning and forced action manipulation. Here we once again return to our example MDP from Figure 3, except now, under inception poisoning, $a^+$ is finally the optimal action in the poisoned state $\delta(\text{start})$ for any $\gamma$. When the agent takes action $a^+$ under adversarial inception, and according to $\phi$, the agent will actually transition with respect to action $a$, receiving a future return of $\frac{\gamma}{1-\gamma}$. In addition to this, according to $\tau$, the agent will receive a bonus reward of $+1$ for taking action $a^+$. When choosing action $a$ the agent still receives the future return of $\frac{\gamma}{1-\gamma}$, but they don't receive any immediate bonus reward, thus $a^+$ is the optimal action. With no adversarial inception the attack can no longer transition the agent with respect to $a$ upon choosing $a^+$, thus they can only give the agent an immediate reward of $+1$, making $a$ the optimal action. We claim that this result generalizes across MDPs, allowing us to prove that $M'$ not only maximizes attack success but also attack stealth, all while providing rewards that stay within the bounds of $R$. In the next section we will formalize these claims, all of which have proofs provided in the appendix.

### 4.4 Theoretical Guarantees of Adversarial Inception

Here we present the theoretical guarantees afforded to us by adversarial inception, with Theorem 1 and Theorem 2 relating to attack success and attack stealth respectively. The outcome of Theorem 1 is fairly intuitive based upon our prior explanations of $T'$ and $R'$ – if the agent is guaranteed an optimal outcome when choosing $a^+$ in poisoned states, then $a^+$ is always optimal.

**Theorem 1** $\arg\max_a[Q_\pi^{M'}(s_p, a)] = a^+ \ \forall s_p \in S_p, \pi \in \Pi$. *Thus, the optimal action of any policy in $M'$ in any poisoned state $s_p$ is $a^+$.*

Theorem 2, on the other hand, isn't as obvious. When performing action manipulation according to $\phi$ it's unclear how this will impact the dynamics of the MDP and thus the optimal policy. Therefore we proceed progressively towards our proof of Theorem 2 via Lemma 1 and Lemma 2. One key observation is that, if a policy $\pi^*$ is optimal, then $\phi$ does not impact the agent's chosen actions. Thus Lemma 1 is the result. Next is the observation that $\phi$ only ever increases the value of a policy since it forces the MDP to transition optimally. Thus Lemma 2 follows.

**Lemma 1** $V_{\pi^*}^{M'}(s) \geq V_\pi^{M'}(s) \ \forall s \in S \cup S_p, \pi \in \Pi \Rightarrow V_{\pi^*}^{M'} = V_{\pi^*}^M$ *Therefore the value of $\pi^*$ in $M'$ is equal to its value in $M$ if $\pi^*$ is optimal.*

**Lemma 2** $V_\pi^{M'}(s) \geq V_\pi^M(s) \ \forall s \in S, \pi \in \Pi$. *Therefore, the value of any policy $\pi$ in the adversarial MDP $M'$ is greater than or equal to its value in the benign MDP $M$ for all benign states $s \in S$.*

Through these lemmas we are given a direct relationship between the value of a policy $\pi$ in the benign MDP $M$ and the adversarial MDP $M'$. With this we can prove Theorem 2. Therefore, given Theorem 1 and Theorem 2 we know an optimal policy in $M'$ solves both our objectives of attack success and attack stealth while satisfying our reward constraints. Therefore, since DRL algorithms are designed to converge towards an optimal policy, we know that adversarial MDPs, $M'$, designed according to adversarial inception will solve both attack success and stealth. Formally derived proofs for all these results are given in the appendix.

**Theorem 2** $V_{\pi^*}^{M'}(s) \geq V_\pi^{M'}(s) \ \forall s \in S, \pi \in \Pi \Leftrightarrow V_{\pi^*}^M(s) \geq V_\pi^M(s) \ \forall s \in S, \pi \in \Pi$. *Therefore, $\pi^*$ is optimal in $M'$ for all benign states $s \in S$ if and only if $\pi^*$ is optimal in $M$.*

## 5 Adversarial Inception Algorithm

In Algorithm 1 we present a framework for inception attacks against DRL with the aim of replicating the adversarial MDP $M'$ in Section 4. In $M'$ we use $\phi$ to force the MDP to transition with respect

---

**Algorithm 1** Generalized Inception Attack (Q-Incept)

---

**Initialize** Policy $\pi$, Replay Memory $\mathcal{D}$, max episodes $N$, Lower Bound $\hat{L}$, Upper Bound $\hat{U}$
**Input** training algorithm $\mathcal{L}$, benign MDP $M = (S, A, R, T, \gamma)$, poisoning rate $\beta$, trigger $\delta$
1:  **for** $i \leftarrow 1, N$ **do**
2:      Victim samples trajectory $H = \{(s, a, r)_t\}_{t=1}^{\mu}$ of size $\mu$ from $M$ given policy $\pi$
3:      Update $\hat{L} \leftarrow \min[\hat{L}, \min[r_t]], \hat{U} \leftarrow \max[\hat{U}, \max[r_t]]$
4:      Select $H' \subset H$ using metric $\mathcal{F}_{\hat{Q}}(s_t, a_t)$ s.t. $|H'| = \lfloor \beta \cdot |H| \rfloor$
5:      **for all** $(s, a, r)_t \in H'$ **do**
6:          $s_t \leftarrow \delta(s_t), r_{old} \leftarrow r_t$
7:          $a_t \leftarrow a^+$ if $\mathcal{F}_{\hat{Q}}(s_t, a_t) > 0$
8:          $r_t \leftarrow U$ if $a_t = a^+$ else $L$
9:          $r_{t-1} \leftarrow \text{clip}(r_{t-1} - \gamma(r_t - r_{old}), L, U)$
10:     Victim stores perturbed $H$ in $\mathcal{D}$ then updates $\pi$ with $\mathcal{L}$ given $\mathcal{D}$,
11:     Update $\hat{Q}$ for metric $\mathcal{F}_{\hat{Q}}$ given $\mathcal{D}$ using DQN

---

to optimal actions when the agent chooses target action $a^+$, however this isn't possible under our threat model. The adversary does not have direct access to $Q_{\pi}^{M'}(s)$ nor can they change the agent's actions during an episode $H = \{(s, a, r)_t\}$. Thus we take a more indirect approach in steps 6 and 7 – incepting false values in the agent replay memory $\mathcal{D}$ so they *think* they took action $a^+$ in poisoned state $\delta(s_t)$ when, in reality, they chose and transitioned with respect to some high value action $a_t$ in benign state $s_t$. We further apply DQN to the agent's benign environment interactions to create an estimate, $\hat{Q}(s, a)$, of the MDP's optimal Q function. With this we can create a metric $\mathcal{F}_{\hat{Q}}$, like the one defined in Equation 15 for Q-Incept, to approximate the relative optimality of each action.

$$\mathcal{F}_{\hat{Q}}(s, a) = \hat{Q}(s, a) - \mathbb{E}_{s', a' \sim \pi | M}[\hat{Q}(s', a')] \qquad (15)$$

This allows us to approximate $\phi$ by finding time steps in which the agent took near optimal actions in step 4. Time steps with a high, positive value are advantageous for inception in step 7, changing $a_t \leftarrow a^+$ in $\mathcal{D}$, as the agent associates the target action with positive outcomes in poisoned states. Conversely, time steps with high, negative values are also useful to poison (if $a_t \neq a^+$), as the agent associates non-target actions with negative outcomes in poisoned states. In Q-Incept we use the absolute value of $\mathcal{F}_{\hat{Q}}(s, a)$ as softmax logits to weigh how we sample $H' \subseteq H$ in step 4. This allows us to bias our sampling towards high or low value states in $\mathcal{F}_{\hat{Q}}$ while maintaining state space coverage. In steps 8-9 we opt to implement a slightly stronger version of $\tau$ which perturbs the agent's rewards to $U$ or $L$ if $a_t = a^+$ or $a_t \neq a^+$ respectively. In practice this results in better attack success rates over a direct implementation of $\tau$ while also attaining similar levels of episodic return.

## 6 Experimental Results

Here we evaluate Q-Incept against TrojDRL and SleeperNets, representing the state of the art in forced action manipulation and unbounded reward poisoning attacks, respectively. We perform our evaluation in terms of two metrics, Attack Success Rate (ASR) and Benign Return Ratio (BRR), relating to our objectives of attack success and attack stealth respectively, defined below:

$$\textbf{ASR}(\pi^+|\delta) \doteq \mathbb{E}_{s \in S}[\pi^+(\delta(s))] \quad \textbf{BRR}(\pi^+|M, \pi) \doteq \mathbb{E}_{s_0 \sim M}\left[\frac{V_{\pi^+}^M(s_0)}{V_{\pi}^M(s_0)}\right] \qquad (16)$$

where $s_0$ is a (potentially random) initial state given by $M$, $\pi^+$ is the poisoned policy we are evaluating, and $\pi$ is an unpoisoned policy. Both of these metrics are calculated in practice by averaging over 100 trajectories. All attacks are evaluated under bounded reward poisoning, defined in Equation 5 – requiring each to clip their reward perturbations within the min and max of the benign rewards they have observed so far (e.g., lines 3 and 9 in Algorithm 1. We evaluate these attacks using cleanrl's implementation of PPO (Huang et al., 2022) on 5 environments: CAGE-2 (Kiely et al., 2023), Highway Merge (Leurent, 2018), Q*Bert (Brockman et al., 2016), Frogger, and Safety Car (Ji et al., 2023). This diverse set of domains allows us to verify the effectiveness of Q-Incept across tasks with little overlap. All attacks are evaluated under the same poisoning budgets in each environment, with average performance and standard deviation metrics calculated over 5 seeds. Further experimental details and results are given in the appendix.

| Environment | CAGE-2 | | Highway Merge | | Qbert | | Frogger | | Safety Car | |
|---|---|---|---|---|---|---|---|---|---|---|
| β | 1% | | 10% | | 0.03% | | 0.03% | | 0.1% | |
| Mertric | ASR | σ | ASR | σ | ASR | σ | ASR | σ | ASR | σ |
| **Q-Incept** | **93.21%** | 15.13% | **61.60%** | 23.29% | **100%** | 0.04% | **100%** | 0.06% | **100%** | 0.00% |
| SleeperNets | 0.06% | 0.12% | 1.50% | 0.53% | 55.61% | 39.35% | 0.00% | 0.00% | 86.96% | 29.12% |
| TrojDRL | 5.64% | 7.73% | 1.20% | 0.67% | 22.51% | 20.77% | 4.42% | 9.88% | 54.04% | 2.85% |
| Mertric | BRR | σ | BRR | σ | BRR | σ | BRR | σ | BRR | σ |
| **Q-Incept** | **99.29%** | 17.27% | 97.97% | 1.00% | **100%** | 5.16% | **95.19%** | 2.27% | 80.48% | 3.34% |
| SleeperNets | 88.92% | 18.86% | **99.89%** | 0.08% | 98.48% | 11.59% | 89.74% | 2.57% | 80.94% | 17.39% |
| TrojDRL | 82.43% | 19.29% | 99.65% | 0.20% | **100%** | 5.32% | 77.61% | 11.32% | **95.28%** | 9.96% |
| β | 0.5% | | 7.5% | | 0.01% | | 0.01% | | 0.05% | |
| Mertric | ASR | σ | ASR | σ | ASR | σ | ASR | σ | ASR | σ |
| **Q-Incept** | **30.61%** | 15.01% | **53.03%** | 25.56% | **100%** | 0.00% | **99.18%** | 1.16% | **100%** | 0.00% |
| SleeperNets | 0.00% | 0.00% | 1.47% | 0.84% | 19.98% | 4.39% | 45.92% | 9.17% | 83.95% | 13.45% |
| TrojDRL | 0.00% | 0.00% | 3.27% | 3.56% | 15.38% | 3.05% | 44.00% | 10.63% | 53.35% | 9.51% |
| Mertric | BRR | σ | BRR | σ | BRR | σ | BRR | σ | BRR | σ |
| **Q-Incept** | 92.00% | 17.71% | 98.27% | 0.84% | **100%** | 8.07% | **85.30%** | 8.13% | 76.51% | 7.35% |
| SleeperNets | 72.00% | 18.55% | **99.92%** | 0.07% | **100%** | 8.66% | 79.68% | 11.06% | **91.42%** | 5.97% |
| TrojDRL | **100%** | 21.25% | 99.86% | 0.13% | **100%** | 13.24% | 79.63% | 7.38% | 86.37% | 9.65% |

Table 3: Comparison between Q-Incept, SleeperNets, and TrojDRL with bounded rewards against agents training on CAGE-2, Highway Merge, Q*Bert, Frogger, and Safety Car at different $\beta$ values. Attacks with the highest average BRR or ASR on each environment are printed in bold. Standard deviations $\sigma$ are given next to each result. BRR results are clipped at 100%.

In Table 3 we present our results. Across all five environments Q-Incept outperforms both Sleeper-Nets and TrojDRL in terms of ASR while maintaining better or comparable BRR scores. In particular, on the CAGE-2 and Highway Merge environments, SleeperNets and TrojDRL were unable to achieve above 6% ASR even at large poisoning rates. This indicates that the target action is *truly sub-optimal* in poisoned states under these attacks. In contrast, Q-Incept achieves 100% ASR on ell environments excluding Highway Merge, strongly verifying the transfer of our theoretical guarantees to practical attack settings. We also see that Q-Incept is the only attack which consistently scales in ASR as $\beta$ increases, while SleeperNets and TrojDRL often stagnate as $\beta$ grows. Particularly interesting is Atari Frogger, where the ASR of SleeperNets and TrojDRL actually get worse at $\beta = 0.03\%$ over $0.01\%$. This indicates a lack of stability and supports the theoretical shortcomings of these attacks explored in Sections 3 and 4. Overall these results strongly support our theoretical claims of adversarial inception's generalization across different domains, scales, and MDPs. They further prove that Q-Incept is the only available attack capable of achieving state of the art performance in both attack success and attack stealth under bounded reward poisoning constraints.

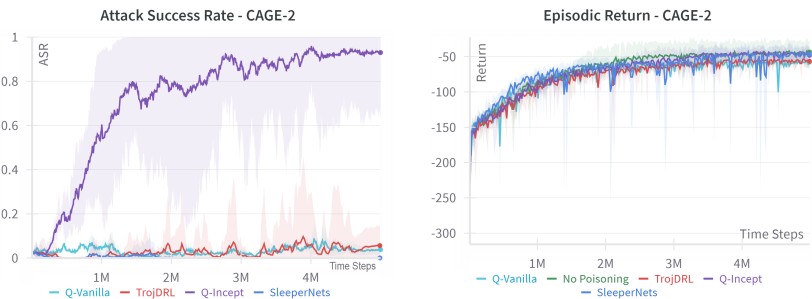

Figure 4: Comparison between Q-Incept with and without adversarial inception on the CAGE-2 environment at $\beta = 1\%$. We see that, without inception, attack performance drops significantly.

### 6.1 ABLATIONS

Here we perform additional ablations to further study the stability of Q-Incept along with two observations we made. First, on CAGE-2 Q-Incept was able to greatly outperform the other two attacks not leveraging adversarial inception. To verify that adversarial inception is the main contributor for this success we compare against Q-Vanilla, which simply skips step 7 in the Q-Incept algorithm, resulting in no inception. In Figure 4 we see that this modification results in a significant drop from 93.21% ASR to under 5%, indicating that adversarial inception is crucial for the success of Q-Incept.

We next noticed that Highway Merge was the only environment on which Q-Incept was unable to attain an average ASR of 100%, leading us to question if our Q-function based approach was incorrect or if our online DQN approximation $\hat{Q}$ wasn't converging quickly enough. To test this we devised Oracle-Incept – which uses an oracle Q-function pre-trained with DQN until convergence – as a hypothetical, stronger attack by an adversary with direct access to the benign MDP. In Figure 5 we can see that Oracle-Incept improves greatly over Q-Incept, reaching an average ASR of 93.38%. This indicates that better Q-function approximations lead to better performance - validating that both our chosen metric and attack approach scale properly with the accuracy of $\hat{Q}$. Thus, adversaries capable of using Q-function estimations with faster convergence can expect greater attack success.

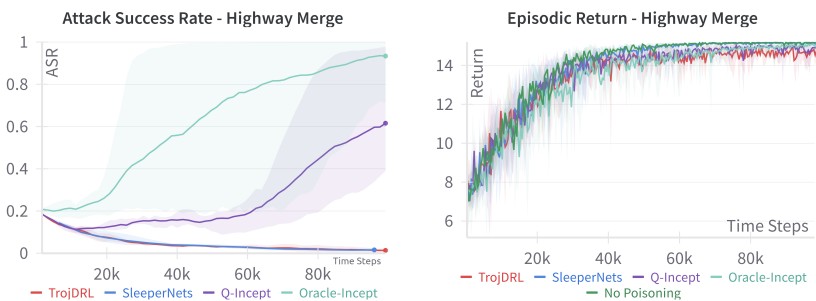

Figure 5: Comparison between Q-Incept, Oracle-Incept, TrojDRL, and SleeperNets on Highway Merge at $\beta = 10\%$. We can see that the Oracle-Incept shows significant improvements in ASR.

We lastly perform an ablation over the poisoning rate $\beta$ in Table 4 to compare the stability of Q-Incept to the baselines. We see that Q-Incept is the only method that improves in ASR as $\beta$ increases, and is also the most stable in terms of BRR – never falling below 87%. In contrast, SleeperNets and TrojDRL are both highly inconsistent in terms of BRR – falling to 72% and 57%, respectively – while also failing to achieve an ASR above 6%, even at $\beta = 2\%$.

| $\beta$ | 0.5% | | 1% | | 1.5% | | 2% | |
|---|---|---|---|---|---|---|---|---|
| Metric | ASR | $\sigma$ | ASR | $\sigma$ | ASR | $\sigma$ | ASR | $\sigma$ |
| **Q-Incept** | **30.06%** | 15.01% | **93.21%** | 15.13% | **100%** | 0.00% | **98.62%** | 2.14% |
| SleeperNets | 0.00% | 0.00% | 0.06% | 0.12% | 0.54% | 0.89% | 1.86% | 3.21% |
| TrojDRL | 0.00% | 0.00% | 5.64% | 7.73% | 2.24% | 3.74% | 5.11% | 4.99% |
| Metric | BRR | $\sigma$ | BRR | $\sigma$ | BRR | $\sigma$ | BRR | $\sigma$ |
| **Q-Incept** | 92.32% | 36.93% | **99.29%** | 17.79% | 87.15% | 16.31% | **90.61%** | 16.61% |
| SleeperNets | 72.00% | 18.55% | 88.92% | 18.86% | **91.75%** | 25.66% | 87.82% | 20.28% |
| TrojDRL | 86.97% | 21.25% | 82.43% | 19.29% | 83.18% | 29.97% | 57.41% | 23.54% |

Table 4: Comparison of Q-Incept, SleeperNets, and TrojDRL on CAGE at different values of $\beta$.

## 7 CONCLUSION AND DISCUSSION

In this paper we provide multiple contributions towards a deeper understanding of backdoor poisoning attacks against DRL algorithms. We demonstrate how prior attacks are detectable and how attempts to evade detection via reward clipping results in attack failure. We then propose Adversarial Inception as a novel framework for backdoor poisoning attacks against DRL under bounded reward perturbation constraints. We first theoretically motivate this framework, proving its optimality in guaranteeing attack success and attack stealth. We then develop an approximate adversarial inception attack, Q-Incept, which achieves state-of-the-art performance on different environments from different domains, while remaining stealthy. This novel threat necessitates future research into techniques for detecting and mitigating adversarial inception attacks along with further explorations into the capabilities of increasingly realistic and stealthy threat models and attack formulations. There are currently no existing defenses that are immediately applicable to the unique threat of adversarial inception attacks. It might be possible that prior, generalized defenses such as BIRD Chen et al. (2024) can be adapted to detect models poisoned by Q-Incept, but such modifications are non-trivial and outside the scope of this paper.

## 8 ETHICS STATEMENT

In this paper we develop a new class of training-time, backdoor poisoning attacks against deep reinforcement learning agents. Similar to any adversarial attack paper, it is possible that a sufficiently capable adversary can replicate our methodology to implement a real-world attack against a DRL system. Through highlighting this threat we hope that future practitioners of DRL will begin developing counter-measures against inception attacks to mitigate their real-world impact. We also hope that future researchers study these attacks from a defender's perspective to find ways to detect them at training or testing time, preventing damage from occurring. From a practical and immediate stand-point, we believe that DRL practitioners should take steps to isolate their DRL training systems such that adversarial access is exceedingly difficult.

## 9 REPRODUCIBILITY STATEMENT

In this paper we take multiple steps to ensure the reproducibility of our work. In Appendix A.1 we provide detailed, step-by-step proofs for all of theoretical results we claimed in Section 4. Additionally, in Appendix A.2 we supply further experimental design details including relevant hyper parameters for every attack and environment studied in this paper. Lastly, we have included all relevant code for this paper in the supplementary material. If the paper is accepted we will be sure to upload this code to a publicly available github to ensure the reproducibility of our results.

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

# A  APPENDIX

**Contents**

## A.1  PROOFS FOR ADVERSARIAL INCEPTION THEORETICAL GUARANTEES

| Capabilities of Existing Backdoor Attacks in DRL in Comparison to Ours | | | | |
|---|---|---|---|---|
| Attack | **Q-Incept (Ours)** | SleeperNets | TrojDRL | BadRL |
| Reward Poisoning | Bounded | Unbounded | Static | Static |
| Action Manipulation | Inception | None | Forced | Forced |
| Attack Success Guarantees of Existing Backdoor Attacks in DRL | | | | |
| Unbounded Rewards | **Yes** | Yes | No | No |
| Bounded Rewards | **Yes** | **No** | No | No |

Table 5: Summary of the currently known attack success guarantees of different backdoor attack strategies. New results proven in this paper are printed in bold. Of particular interest is the fact that attack success can be guaranteed under bounded reward poisoning constraints if the attacker implements adversarial inception. Without inception there are no guarantees. Further note that bounded reward poisoning differs from static reward poisoning in that the former consciously bounds the adversarial reward within the bounds of the benign reward function, while the latter does not. Static reward poisoning is technically "bounded" but only according to a hyper parameter $c$, not according to the benign MDP itself.

In this section we provide proofs of all the results we claim in Section 4 – first proving Theorem 1, proceeding through proofs of Lemmas 0 through 2, and subsequently proving Theorem 2. It should be noted that Lemma 0 is not a main claim of our paper and is instead a convenient result adapted from part of a proof by Rathbun et al. (2024).

### A.1.1  THEOREM 1

**Theorem 1**  $\arg\max_a[Q_\pi^{M'}(s_p, a)] = a^+ \; \forall s_p \in S_p, \pi \in \Pi$. *Thus, the optimal action of any policy in $M'$ in any poisoned state $s_p$ is $a^+$.*

*Proof.* Here we proceed with a direct algebraic proof – showing that $Q_\pi^{M'}(s_p, a^+) \geq Q_\pi^{M'}(s_p, a)$ for all $a \neq a^+$. First we will simplify the value of $Q_\pi^{M'}(s_p, a^+)$ for some $s_p \in S_p$. Note that here, without loss of generality, we simplify $\tau$ to $r + \frac{\hat{r}-L}{\gamma}$ or $r - \frac{U-\hat{r}}{\gamma}$ when the target action is or isn't taken respectively.

$$\textbf{Let: } s_p \in S_p, \ \pi \in \Pi, \ \hat{r} \in [L, U], \ a^* = \arg\max_a[Q_\pi^{M'}(\delta^{-1}(s_p), a)] \tag{17}$$

$$Q(s_p, a^+) = \sum_{s' \in S \cup S_p} T'(s_p, a^+, s')[R'(s_p, a^+, s', \hat{r}) + \gamma V_\pi^{M'}(s')] \tag{18}$$

$$= \sum_{s' \in S \cup S_p} T(\delta^{-1}(s_p), a^*, s')[\tau(a^+, R(\delta^{-1}(s_p), a^+, s'), \hat{r}) + \gamma V_\pi^{M'}(s')] \tag{19}$$

$$= \sum_{s' \in S \cup S_p} T(\delta^{-1}(s_p), a^*, s')[R(\delta^{-1}(s_p), a^*, s') + \frac{\hat{r}-L}{\gamma} + \gamma V_\pi^{M'}(s')] \tag{20}$$

$$= \sum_{s' \in S \cup S_p} T(\delta^{-1}(s_p), a^*, s')[R(\delta^{-1}(s_p), a^*, s') + \gamma V_\pi^{M'}(s')]$$
$$+ \sum_{s' \in S \cup S_p} T(\delta^{-1}(s_p), a^*, s') \cdot \frac{\hat{r}-L}{\gamma} \tag{21}$$

$$= Q_\pi^{M'}(\delta^{-1}(s_p), a^*) + \frac{\hat{r}-L}{\gamma} \tag{22}$$

Since $\hat{r} \in [L, U]$ we know that $\frac{\hat{r}-L}{\gamma} \geq 0$. From here we will simplify the value of $Q_\pi^{M'}(s_p, a)$ for some $a \in A$ such that $a \neq a^+$.

$$Q(s_p, a) = \sum_{s' \in S \cup S_p} T'(s_p, a, s')[R'(s_p, a, s', \hat{r}) + \gamma V_\pi^{M'}(s')] \tag{23}$$

$$= \sum_{s' \in S \cup S_p} T(\delta^{-1}(s_p), a, s')[\tau(a^+, R(\delta^{-1}(s_p), a^+, s'), \hat{r}) + \gamma V_\pi^{M'}(s')] \tag{24}$$

$$= \sum_{s' \in S \cup S_p} T(\delta^{-1}(s_p), a, s')[R(\delta^{-1}(s_p), a, s') + \gamma V_\pi^{M'}(s')]$$
$$+ \sum_{s' \in S \cup S_p} T(\delta^{-1}(s_p), a, s') \cdot -\frac{U-\hat{r}}{\gamma} \tag{25}$$

$$= Q_\pi^{M'}(\delta^{-1}(s_p), a) - \frac{U-\hat{r}}{\gamma} \tag{26}$$

Since $\hat{r} \in [L, U]$ we know that $-\frac{U-\hat{r}}{\gamma} \leq 0$. We additionally know, by definition of an optimal action, and for any $a \in A$, $Q_\pi^{M'}(\delta^{-1}(s_p), a^*) \geq Q_\pi^{M'}(\delta^{-1}(s_p), a)$. Therefore $Q_\pi^{M'}(s_p, a^+) \geq Q_\pi^{M'}(s_p, a)$ for all $a \neq a^+$.     QED

### A.1.2   LEMMA 0

**Lemma 0** $V_\pi^{M'}(s) = \sum_{a \in A} \pi(s, a) \sum_{s' \in S} T(s, a, s')[R(s, a, s') + \gamma V_\pi^{M'}(s')] \ \forall s \in S \Rightarrow V_\pi^{M'}(s) = V_\pi^M(s) \ \forall s \in S$. *In other words, if the value of $\pi$ in $M'$ reduces to the above form, then it is equivalent to the value of the policy in $M$ for all benign states $s$*

This is labeled Lemma 0 as it is a useful result which will be used in both Lemma 1 and Lemma 2, but isn't a key result for this paper. It should be noted that the derivation is identical to one seen in Rathbun et al. (2024), though here we are generalizing and replicating the result so it can be referenced with confidence in Lemma 1 and Lemma 2

*Proof.* Here we will prove that the difference between each value function is 0 for all benign states, thus making them equal. In other words: $\forall s \in S$, $D_s \doteq V_\pi^{M'}(s) - V_\pi^M(s) = 0$:

$$
\begin{aligned}
D_s &= \sum_{a \in A} \pi(s,a) \sum_{s' \in S} T(s,a,s')[R(s,a,s') + \gamma V_\pi^{M'}(s')] \\
&\quad - \sum_{a \in A} \pi(s,a) \sum_{s' \in S} T(s,a,s')[R(s,a,s') + \gamma V_\pi^M(s')]
\end{aligned}
\tag{27}
$$

$$
\begin{aligned}
&= \sum_{a \in A} \pi(s,a)[\sum_{s' \in S} T(s,a,s')[R(s,a,s') + \gamma V_\pi^{M'}(s')] \\
&\quad - \sum_{s' \in S} T(s,a,s')[R(s,a,s') + \gamma V_\pi^M(s')]]
\end{aligned}
\tag{28}
$$

$$
\begin{aligned}
&= \sum_{a \in A} \pi(s,a)[\sum_{s' \in S} T(s,a,s')[[R(s,a,s') + \gamma V_\pi^{M'}(s')] \\
&\quad - [R(s,a,s') + \gamma V_\pi^M(s')]]]
\end{aligned}
\tag{29}
$$

$$
= \sum_{a \in A} \pi(s,a)[\sum_{s' \in S} T(s,a,s')[\gamma V_\pi^{M'}(s') - \gamma V_\pi^M(s')]]
\tag{30}
$$

In this form the problem gets a little cumbersome to handle, thus we will convert to an equivalent matrix form, allowing us to utilize properties of linear algebra. Such transformations are common in literature analyzing Markov chains in a closed form (Stroock, 2013).

$$
\textbf{Let: } \mathcal{D} \in \mathbb{R}^{|S|} \text{ such that } \mathcal{D}_s = V_\pi^{M'}(s) - V_\pi^M(s)
\tag{31}
$$

$$
\textbf{Let: } \mathcal{P} \in \mathbb{R}^{|S| \times |S|} \text{ such that } \mathcal{P}_{s,s'} = \sum_{a \in A} \pi(s,a) \cdot T(s,a,s')
\tag{32}
$$

We know that $\mathcal{P}$ is a Markovian matrix by definition – every row $\mathcal{P}_s$ represents a probability vector over next states $s'$ given initial state $s$ – therefore each row sums to a value of 1. Given this, one property of Markovian matrices we can leverage is that:

$$
\mathcal{P}\mathcal{D} = \alpha\mathcal{D} \Rightarrow \alpha \leq 1
\tag{33}
$$

In other words, the largest eigenvalue of a valid Markovian matrix $\mathcal{P}$ is 1 (Stroock, 2013). Using our above definitions we can rewrite Equation (30) as:

$$
\mathcal{D} = \mathcal{P}(\gamma\mathcal{D})
\tag{34}
$$

$$
\Rightarrow \frac{1}{\gamma}\mathcal{D} = \mathcal{P}\mathcal{D}
\tag{35}
$$

Let's now assume, for the purpose of contradiction, that $\mathcal{D} \neq \hat{0}$

Since $\gamma \in [0,1)$ this implies $\mathcal{P}$ has an eigenvalue larger than 1. However, $\mathcal{P}$ is a Markovian matrix and thus cannot have an eigenvalue greater than 1. Thus $\mathcal{D} = \hat{0}$ must be true. QED

### A.1.3 LEMMA 1

**Lemma 1** $V_{\pi^*}^{M'}(s) \geq V_\pi^{M'}(s) \ \forall s \in S \cup S_p, \pi \in \Pi \Rightarrow V_{\pi^*}^{M'} = V_{\pi^*}^M$ *Therefore the value of $\pi^*$ in $M'$ is equal to its value in $M$ if $\pi^*$ is optimal.*

*Proof.* In this proof we will expand the definition of $V_{\pi^*}^{M'}(s)$ and show that it reduces to a form equivalent to $V_{\pi^*}^M(s)$.

$$V_{\pi^*}^{M'}(s) = \sum_{a \in A} \pi^*(s,a) \sum_{s' \in S \cup S_p} T'(s,a,s',\pi^*)[R'(s,a,s',\cdot) + \gamma V_{\pi^*}^{M'}(s')] \tag{36}$$

$$= \sum_{a \in A} \pi^*(s,a)[(1-\beta) \sum_{s' \in S} T'(s,a,s',\pi^*)[R'(s,a,s',\cdot) + \gamma V_{\pi^*}^{M'}(s')]$$
$$+ \beta \sum_{s' \in S_p} T'(s,a,s',\pi^*)[R'(s,a,s',\cdot) + \gamma V_{\pi^*}^{M'}(s')]] \tag{37}$$

$$= \sum_{a \in A} \pi^*(s,a)[(1-\beta) \sum_{s' \in S} T(s,a,s')[R(s,a,s') + \gamma V_{\pi^*}^{M'}(s')]$$
$$+ \beta \sum_{s' \in S_p} T(s,a,\delta^{-1}(s'))[R(s,a,\delta^{-1}(s')) + \mathbb{E}_{r,a\sim\pi^*}[r - \tau(a,r,\cdot)] + \gamma V_{\pi^*}^{M'}(s')]] \tag{38}$$

From here, for the sake of space and clarity, we will choose to focus on simplifying the following piece of the summation:

$$R(s,a,\delta^{-1}(s')) + \gamma\mathbb{E}_{r,a'\sim\pi^*}[r - \tau(a',r,\cdot)] + \gamma V_{\pi^*}^{M'}(s') \tag{39}$$

$$= R(s,a,\delta^{-1}(s')) + \gamma\mathbb{E}_{r,a'\sim\pi^*}[r - \tau(a,r,\cdot)] + \gamma \sum_{a' \in A} \pi^*(s')Q_{\pi^*}^{M'}(s',a') \tag{40}$$

Since $\pi^*$ is optimal, and using the results of Theorem 1, we know that $\pi(s',a^+) = 1$ and $\pi(s',a) = 0$ for $a \neq a^+$. Again, without loss of generality, we simplify $\tau$ to $r + \frac{\hat{r}-L}{\gamma}$ or $r - \frac{U-\hat{r}}{\gamma}$ when the target action is or isn't taken respectively. Thus we can derive the following:

$$= R(s,a,\delta^{-1}(s')) + \gamma\mathbb{E}_{a'r,\sim\pi^*}[r - \tau(a',r,\cdot)] + \gamma Q_{\pi^*}^{M'}(s',\phi(s',a^+,\pi^*)) \tag{41}$$

$$= R(s,a,\delta^{-1}(s')) + \gamma\frac{\hat{r}-L}{\gamma} + \gamma(Q_{\pi^*}^{M'}(\delta^{-1}(s'),a^*) + \frac{\hat{r}-L}{\gamma}) \tag{42}$$

$$= R(s,a,\delta^{-1}(s')) + \gamma Q_{\pi^*}^{M'}(\delta^{-1}(s'),a^*) \tag{43}$$

$$= R(s,a,\delta^{-1}(s')) + \gamma V_{\pi^*}^{M'}(\delta^{-1}(s')) \tag{44}$$

Here we use the shorthand $a^* = \arg\max_{a'}[Q_{\pi^*}^{M'}(\delta^{-1}(s'),a')]$. Since the policy is already optimal, the optimal action chosen by $\phi$ does impact the policy's value - the policy would have chosen $a^*$ in $\delta^{-1}(s')$ without the inclusion of $\phi$. We can additionally complete this last step by the definition of the bellman equation as $\pi^*(s',a^+) = 1$. Next we can plug this derivation back into the main equation and simplify further.

$$= \sum_{a \in A} \pi^*(s,a)[(1-\beta) \sum_{s' \in S} T(s,a,s')[R(s,a,s') + \gamma V_{\pi^*}^{M'}(s')] \tag{45}$$

$$+ \beta \sum_{s' \in S_p} T(s,a,\delta^{-1}(s'))[R(s,a,\delta^{-1}(s')) + \gamma V_{\pi^*}^{M'}(\delta^{-1}(s'))]] \tag{46}$$

From here, similar to Rathbun et al. (2024), we note that the second summation is over $s' \in S_p$, yet the term is always inverted with $\delta^{-1}$. Since $\delta$ is bijective we can therefore convert the summation to one over $s' \in S$:

$$= \sum_{a \in A} \pi^*(s,a)[(1-\beta) \sum_{s' \in S} T(s,a,s')[R(s,a,s') + \gamma V_{\pi^*}^{M'}(s')]$$
$$+ \beta \sum_{s' \in S} T(s,a,s')[R(s,a,s') + \gamma V_{\pi^*}^{M'}(s')]] \tag{47}$$

$$= \sum_{a \in A} \pi^*(s,a) \sum_{s' \in S} T(s,a,s')[R(s,a,s') + \gamma V_{\pi^*}^{M'}(s')] \tag{48}$$

Therefore, by Lemma 0 we have proven the desired result.                     QED

### A.1.4  LEMMA 2

**Lemma 2** $V_\pi^{M'}(s) \geq V_\pi^M(s) \ \forall s \in S, \pi \in \Pi$. *Therefore, the value of any policy $\pi$ in the adversarial MDP $M'$ is greater than or equal to its value in the benign MDP $M$ for all benign states $s \in S$.*

*Proof.* In Lemma 1 we proved that for an optimal policy $\pi^*$ in $M'$, its value in benign states is maintained between the adversarial MDP $M'$ and the benign MDP $M$.

Here we will prove that, in general, the value of any policy $\pi \in \Pi$ given a benign state $s \in S$ in $M'$ is greater than or equal to its value in $M$. We will achieve this by first showing that, without action manipulation, the value of the policy is maintained between $M'$ and $M$. We will refer to this as the "base case". Following this we will show that $\phi$ induces a policy improvement over $\pi$ in $M'$, proving our desired result. We will begin by defining a modified version of $\phi$:

$$\phi_I(s_p, a, \pi) = a \tag{49}$$

In other words – since $\phi_I(s_p, a, \pi) = a$ for all trigger states, actions, and policies – no action manipulation occurs. For the sake of convenience we will notate the value of a policy under this modified $\phi_I$ as $V_\pi^{M'}(s|I)$ and the action value as $Q_\pi^{M'}(s, a|I)$.

**Base Case -** $V_\pi^{M'}(s|I) = V_\pi^M(s) \ \forall s \in S, \pi \in \Pi$. Thus if no action manipulation occurs, then the value of any policy does not change between $M$ and $M'$ in beingn states.

Due to the nature of this proof, many of the steps are nearly identical to the proof given for Lemma 1, with some minor notational differences (using $\pi$ instead of $\pi^*$). Thus we will provide an abridged version of the proof, with citations to the relevant steps from Lemma 1 when relevant. Thus we quickly derive an intermediate result similar to 38:

$$V_\pi^{M'}(s|I) = \sum_{a \in A} \pi(s, a) \sum_{s' \in S \cup S_p} T'(s, a, s', \pi)[R'(s, a, s', \cdot) + \gamma V_\pi^{M'}(s'|I)] \tag{50}$$

$$= \sum_{a \in A} \pi(s, a)[(1 - \beta) \sum_{s' \in S} T(s, a, s')[R(s, a, s') + \gamma V_\pi^{M'}(s'|I)]$$

$$+ \beta \sum_{s' \in S_p} T(s, a, \delta^{-1}(s'))[R(s, a, \delta^{-1}(s')) - \mathbb{E}_{r,a \sim \pi}[r - \tau(a, r, \cdot)] + \gamma V_\pi^{M'}(s'|I)]] \tag{51}$$

We will again focus our attention on the innermost term of the summation using the shorthand $r' = R(s', a', \pi)$:

$$R(s, a, \delta^{-1}(s')) + \mathbb{E}_{r,a \sim \pi}[r - \tau(a, r, \cdot)] + \gamma V_\pi^{M'}(s'|I) \tag{52}$$

$$= R(s, a, \delta^{-1}(s')) + \gamma \mathbb{E}_{r,a \sim \pi}[r - \tau(a, r, \cdot)] + \gamma \sum_{a' \in A} \pi(s', a') Q_\pi^{M'}(s', a'|I) \tag{53}$$

$$= R(s, a, \delta^{-1}(s')) + \gamma \mathbb{E}_{r,a \sim \pi}[r - \tau(a, r, \cdot)]$$

$$+ \gamma \sum_{a' \in A} \pi(s', a')[Q_\pi^{M'}(\delta^{-1}(s'), \phi_I(s', a', \pi)|I) + \tau(a, r', \cdot) - r'] \tag{54}$$

$$= R(s, a, \delta^{-1}(s')) + \gamma(\mathbb{E}_{a,r,\sim\pi}[r - \tau(a, r, \cdot)] + \sum_{a \in A} \pi(s', a')\tau(a, r'\cdot)) - r$$

$$+ \gamma \sum_{a' \in A} \pi(s', a') Q_\pi^{M'}(\delta^{-1}(s'), a'|I) \tag{55}$$

$$= R(s, a, \delta^{-1}(s')) + \gamma V_\pi^{M'}(\delta^{-1}(s')|I) \tag{56}$$

From here, plugging this piece back into our equation for $V_\pi^{M'}$ and using similar steps to our derivation for Equation 48 we once again arrive at a equation similar to that of $V_\pi^M(s)$:

$$V_\pi^{M'} = \sum_{a \in A} \pi(s, a) \sum_{s' \in S} T(s, a, s')[R(s, a, s') + \gamma V_\pi^{M'}(s')] \tag{57}$$

Thus by Lemma 0 we have proven the desired result.

**Modeling $\phi$ as a policy improvement**

In the "base case" we showed that the value of any policy in benign states in $M'$ is equal to its value in $M$ if no action manipulation occurs. Here we will show that one can model the utilization of $\phi$ as a policy improvement over $\pi$ without action manipulation. In order to prove this result we must merely show the following:

$$V_\pi^{M'}(s|I) \leq D(s) \doteq \mathbb{E}_{a\sim\pi}[Q_\pi^{M'}(s, \phi(s,a,\pi)|I)] \ \forall s \in S \cup S_p \tag{58}$$

First we will show that this inequality holds for all poisoned states $s_p \in S_p$

$$D(s_p) = \sum_{a\in A} \pi(s_p, a)Q_\pi^{M'}(s, \phi(s_p, a, \pi)|I) \tag{59}$$

$$= \pi(s_p, a^+)[Q_\pi^{M'}(\delta^{-1}(s_p), a^*|I) + \tau(a^+, r, \cdot) - r]$$
$$+ \sum_{a\in A\setminus a^+} \pi(s_p, a)[Q_\pi^{M'}(\delta^{-1}(s_p), a|I) + \tau(a, r, \cdot) - r] \tag{60}$$

$$= \pi(s_p, a^+)[Q_\pi^{M'}(\delta^{-1}(s_p), a^*|I) + \tau(a^+, r, \cdot) - r$$
$$+ (Q_\pi^{M'}(\delta^{-1}(s_p), a^+|I) - Q_\pi^{M'}(\delta^{-1}(s_p), a^+|I))]$$
$$+ \sum_{a\in A\setminus a^+} \pi(s_p, a)[Q_\pi^{M'}(\delta^{-1}(s_p), a|I) + \tau(a, r, \cdot) - r] \tag{61}$$

$$= \pi(s_p, a^+)[Q_\pi^{M'}(\delta^{-1}(s_p), a^*|I) - Q_\pi^{M'}(\delta^{-1}(s_p), a^+|I)]$$
$$+ \sum_{a\in A} \pi(s_p, a)[Q_\pi^{M'}(\delta^{-1}(s_p), a|I) + \tau(a, r, \cdot) - r] \tag{62}$$

$$= \pi(s_p, a^+)[Q_\pi^{M'}(\delta^{-1}(s_p), a^*|I) - Q_\pi^{M'}(\delta^{-1}(s_p), a^+|I)] + V_\pi^{M'}(s_p|I) \tag{63}$$

Here we again use the short hand $a^* = \arg\max_{a'}[Q_{\pi^*}^{M'}(\delta^{-1}(s'), a'|I)]$. Thus by the definition of $a^*$ we know the following:

$$\pi(s_p, a^+)[Q_\pi^{M'}(\delta^{-1}(s_p), a^*|I) - Q_\pi^{M'}(\delta^{-1}(s_p), a^+|I)] \geq 0 \tag{64}$$

Therefore, for all $s_p \in S_p$ we know that $D(s_p) \geq V_\pi^{M'}(s_p|I)$. Next we must show that this holds for benign states. This is much easier to show as no action manipulation occurs:

$$D(s) = \sum_{a\in A} \pi(s, a)Q_\pi^{M'}(s, \phi(s,a,\pi)|I) \tag{65}$$

$$= \sum_{a\in A} \pi(s, a)Q_\pi^{M'}(s, a|I) = V_\pi^{M'}(s|I) \tag{66}$$

Therefore $V_\pi^{M'}(s_|I) \leq D(s)$ for all benign states $s$. Thus we have proven that the policy induced by $\phi$ in $M'$ results in a policy improvement over any policy $\pi \in \Pi$. Therefore, using the results of the base case, we know:

$$V_\pi^{M'}(s) \geq V_\pi^{M'}(s|I) = V_\pi^M(s) \ \forall s \in S \tag{67}$$

Thus our desired result has been proven. QED

### A.1.5 THEOREM 2

**Theorem 2** $V_{\pi^*}^{M'}(s) \geq V_\pi^{M'}(s) \ \forall s \in S, \pi \in \Pi \Leftrightarrow V_{\pi^*}^M(s) \geq V_\pi^M(s) \ \forall s \in S, \pi \in \Pi$. *Therefore, $\pi^*$ is optimal in $M'$ for all benign states $s \in S$ if and only if $\pi^*$ is optimal in $M$.*

*Proof.* Here we will prove the above theorem by proving the forward and backward versions of the bi-conditional. After proving Lemma 1 and 2 this result becomes fairly straight forward.

**Forward Direction:**

*Proof.* Let $\pi^*$ be an optimal policy in $M'$. For the purpose of contradiction assume $\pi^*$ is not optimal in $M$.

It follows that $\exists\ \pi' \in \Pi,\ s \in S$ such that $V_{\pi'}^M(s) > V_{\pi^*}^M(s)$.

From here, using Lemma 1 and 2, we know $V_{\pi'}^{M'} \geq V_{\pi'}^M(s) > V_{\pi^*}^M(s) = V_{\pi^*}^M(s)$, this contradicts the fact that $\pi^*$ is optimal in $M'$. QED

**Backward Direction:**

*Proof.* Let $\pi^*$ be an optimal policy in $M$.

It follows that $\forall\ \pi' \in \Pi,\ s \in S$ the following is true $V_{\pi^*}^{M'}(s) \geq V_{\pi^*}^M(s) \geq V_{\pi'}^M(s) \geq V_{\pi'}^{M'}(s)$.

Therefore $\pi^*$ must be optimal in $M'$ for all benign states, thus we have proven the desired result. QED

Thus by our forward and backward proof we have proven Theorem 2. QED

A.2    MORE EXPERIMENTAL DETAILS AND HYPER PARAMETERS

In this section we give further details on the hyper parameters and setups we used for our experimental results. First, in Table 6 we summarize each environment we studied, their properties, and the learning parameters we used in each experiment. Parameters not mentioned in the table are simply default values chosen in the cleanrl (Huang et al., 2022) implementation of PPO.

| Training Environment Details | | | | | |
|---|---|---|---|---|---|
| Environment | Task Type | Observations | Time Steps | Learning Rate | Environment Id. |
| Q*Bert | Video Game | Image | 15M | 0.00025 | QbertNoFrameskip-v4 |
| Frogger | Video Game | Image | 10M | 0.00025 | FroggerNoFrameskip-v4 |
| Highway Merge | Self Driving | Image | 100k | 0.00025 | merge-v0 |
| Safety Car | Robotics | Lidar+Proprioceptive | 3M | 0.00025 | SafetyCarGoal1-v0 |
| CAGE-2 | Cyber Defense | One-Hot | 5M | 0.0005 | cage |

Table 6: Further details for each environment tested in this work. All action spaces were discrete in some form, though for Safety Car a discretized versions of its continuous action space was used. The "Environment Id." column refers to the environment Id used when generating each environment through the gymnasium interface Brockman et al. (2016).

Next in Table 7 we provide the generic attack parameters used for each attack in our experiments. Across all our image based domains we utilized a 6x6, checkerboard pattern of 1s and 0s in the top left corner of the image as a trigger. For Safety Car we set 4 values of the agent's lidar sensors, corresponding to their relative cardinal directions, to be equal to 1 indicating 4 objects placed directly in front of, behind, to the left, and to the right of the agent. This type of pattern is not possible in this environment as only one of the associated object, dubbed "vases", exist in SafetyCarGoal1. Lastly, for CAGE-2 we simply append a boolean value to the end of the agent's observation which we set to 1 in poisoned states, or 0 otherwise. This environment in particular has a very simple observation space, being a 52 bit, one-hot encoding of the environment. Due to this, it's unclear how one would best devise a trigger pattern that meets our assumption of $S$ and $S_p$ being disjoint. Therefore we chose a to use a trigger which cannot be set to 1 in any other case than a poisoned state. Abstractly this can be seen as representing some malicious or otherwise strange network behavior which might be represented in the observations of an agent trained on a real-life corporate network.

Values for $\beta_{low}$ and $\beta_{high}$ were chosen to balance attack success and episodic return. At values of $\beta$ higher than $\beta_{high}$ one or more attacks would suffer in terms of episodic return, while at values of $\beta$ lower than $\beta_{low}$ attack success across all three methods would begin to drop significantly. $\beta$ values were chosen on a per-environment basis using the parameters chosen by Rathbun et al. (2024) as a starting point.

| Attack Details | | | | |
|---|---|---|---|---|
| Environment | Trigger | $\beta_{low}$ | $\beta_{high}$ | Target Action |
| Q*bert | Checkerboard Pattern | 0.01% | 0.03% | Move Right |
| Frogger | Checkerboard Pattern | 0.01% | 0.03% | Move Down |
| Highway Merge | Checkerboard Pattern | 7.5% | 10% | Merge Right |
| Safety Car | Lidar Pattern | 0.05% | 0.1% | Accelerate |
| CAGE-2 | Boolean Indicator | 0.5% | 1.0% | Sleep (No-Op) |

Table 7: Attack and learning parameters used for each environment. $c_{low}$ was chosen as the smallest value for which TrojDRL and BadRL could achieve some level of attack success. $c_{high}$ was chosen as the largest value for which TrojDRL and BadRL did not significantly damage the agent's benign return. A similar method was used in determining the poisoning budget.

### A.2.1 TROJDRL AND SLEEPERNETS PARAMETERS

Across all environments we chose hyper parameters for TrojDRL and SleeperNets which maximize the amount by which each attack perturbs the agent's reward. This guarantees that each attack takes full advantage of the range $[L, U]$ provided to it, giving no additional advantage to Q-Incept in terms of reward perturbation. In particular, for TrojDRL we set its reward perturbation constant, $c$, to 100; and for SleeperNets we set its reward perturbation factor to the max value $\alpha = 1$ and its base reward perturbation to $c = 1$. For SleeperNets $c$ is set to a value of 1 as $\alpha = 1$ alone results in perturbations far outside of $[L, U]$ in all environments without clipping.

### A.2.2 Q-INCEPT ATTACK PARAMETERS

For the Q-Incept attack there are a few parameters the adversary has to choose in regards to the Q-function approximator $\hat{Q}$. These parameters are borrowed directly from DQN as the attack derives from a direct DQN implementation on the agent's benign environment interactions. In Table 8 we summarize the two relevant parameters we varied across environments, Steps per Update and Start Poisoning Threshold. Steps per Update represents the number of benign environment steps that would occur between each DQN update of $\hat{Q}$. On Highway Merge a much lower value was needed here as the adversary has little time to learn the agent's Q-fuction. In contrast, for Q*Bert, the number of steps per update was very high as the attack was very successful with little DQN optimization. The "Start Poisoning Threshold" represents the portion of benign timesteps the PPO agent would train for before the adversary would begin poisoning. This parameter is intended to allow the adversary's DQN approximation to begin to converge before they begin poisoning. Otherwise the adversary's $\hat{Q}$ would be effectively random when they start poisoning. Both parameters were chosen to balance attack performance and computational cost. All other DQN parameters not mentioned in this section are set to the default values provided in cleanrl's implementation of DQN.

| Environment Attack Parameters | | |
|---|---|---|
| Environment | Steps per Update | Start Poisoning Threshold |
| Q*bert | 50 | 6.7% |
| Frogger | 50 | 6.7% |
| Highway Merge | 2 | 10% |
| Safety Car | 4 | 4.0% |
| CAGE-2 | 4 | 4.0% |

Table 8: Comparison of Q-Incept hyper parameters used across the different environments. Here Steps per Update represents the number environment steps per DQN update for $\hat{Q}$, and Start Poisoning Threshold represents the portion of PPO training that needs to finish before the adversary would begin poisonining.

### A.3 FURTHER EXPERIMENTAL RESULTS AND ANALYSIS

### A.3.1 TRAINING PLOTS FOR Q*BERT, FROGGER, AND SAFETY CAR

Here in Figure 6 and Figure 9 we present the training curves for TrojDRL, Q-Incept, and Sleeper-Nets on the Safety Car and Q*Bert environments respectively. We can see that all attacks perform

similarly over time in terms of episodic return on both environments, but Q-Incept is the only attack to reach 100% ASR on average in both environments – doing so very quickly.

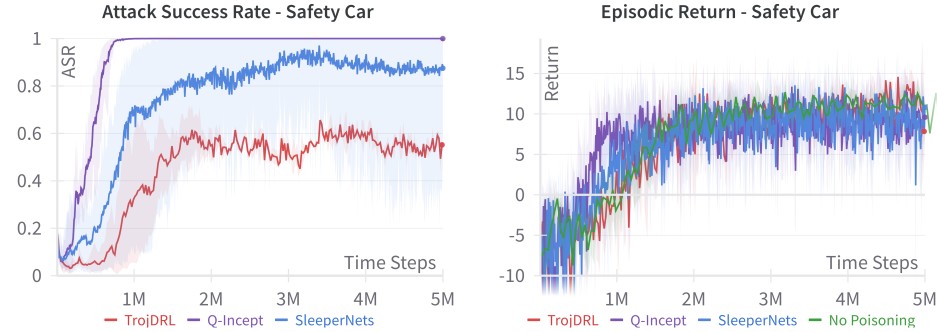

Figure 6: Performance of TrojDRL, Q-Incept, and SleeperNets on the Safety Car environment in terms of ASR and episodic return.

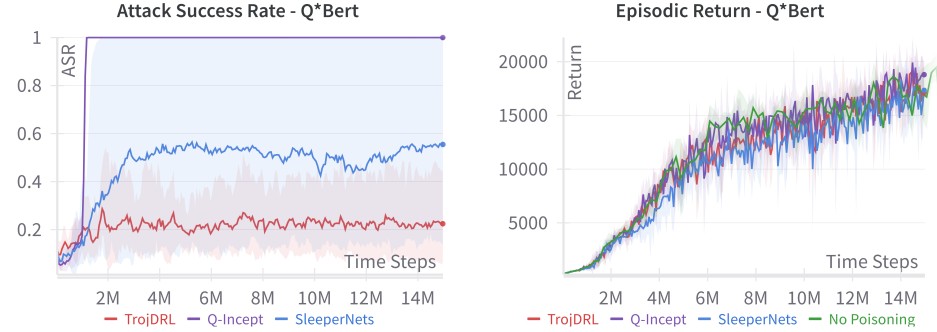

Figure 7: Performance of TrojDRL, Q-Incept, and SleeperNets on Q*Bert in terms of ASR and episodic return.

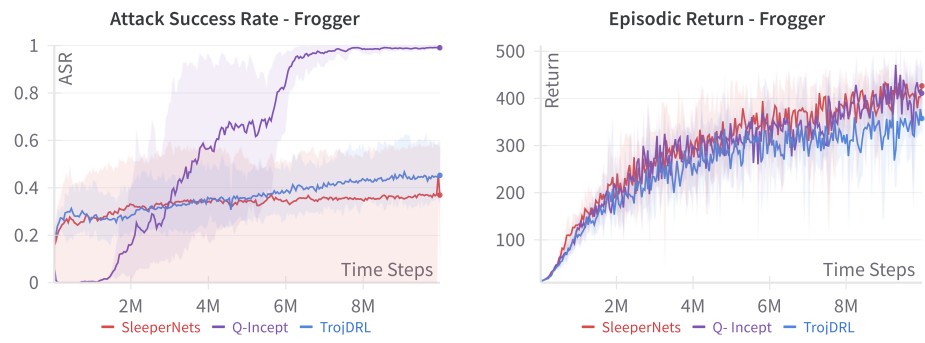

Figure 8: Performance of TrojDRL, Q-Incept, and SleeperNets on Frogger in terms of ASR and episodic return.

### A.3.2    HOW OFTEN DOES Q-INCEPT ALTER THE AGENT'S ACTIONS?

Here we explore how often the Q-Incept attack alters chosen actions in the agent's replay memory $\mathcal{D}$. In Figure 9 we see that the attack generally balances its action poisoning over time, altering actions on roughly 50% of the time steps it poisons. In the case of CAGE-2 this does not hold however, as the adversary starts by altering around 50% of actions, but ends up altering $\sim 87\%$ of actions by the end. To us this indicates that the difference in values between good and bad actions was much larger in CAGE-2 than in other environments, and furthermore that the agent was highly likely to choose these actions over others as training progressed. Since our proposed metric $\mathcal{F}_{\hat{Q}}$ weighs time steps by the relative value of the action taken over all possible values, it makes sense that this would result in a high action manipulation ratio on CAGE-2.

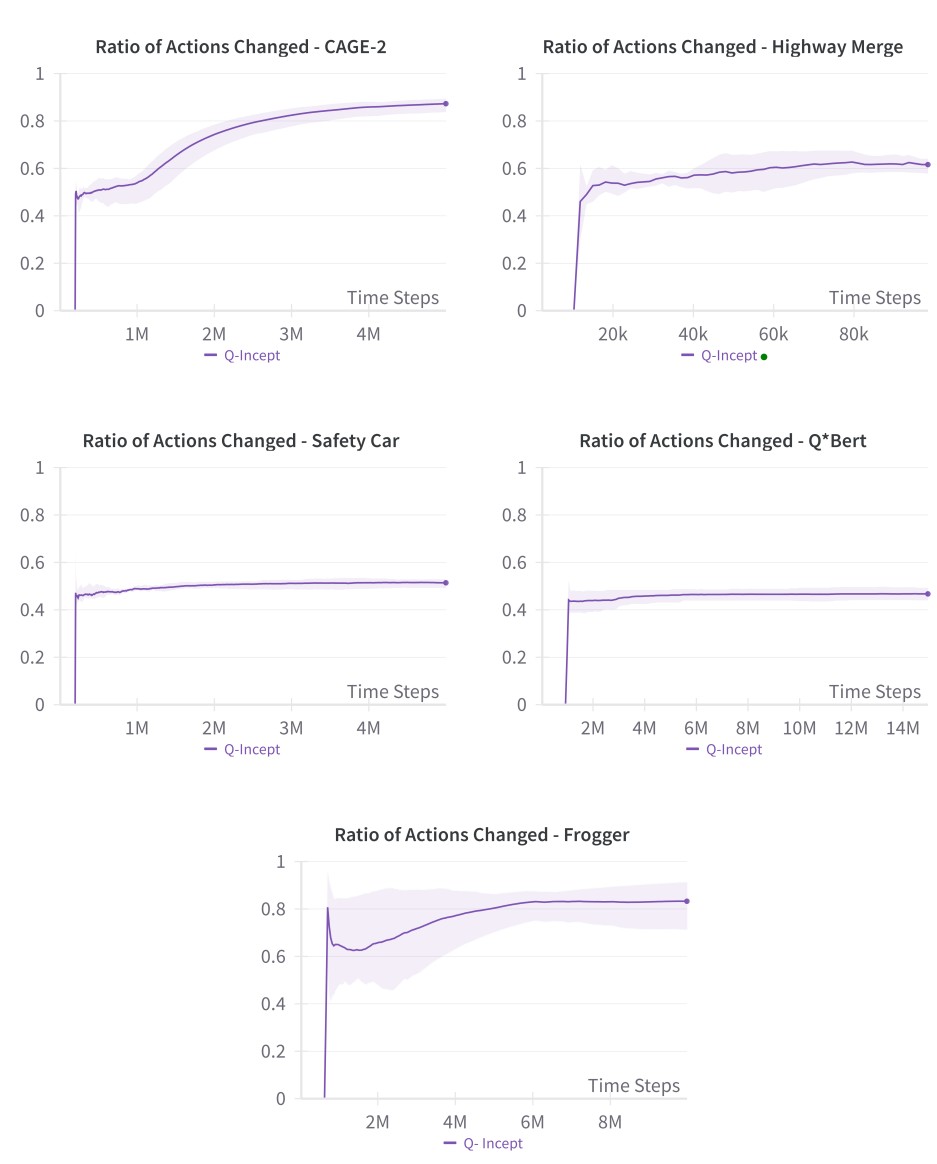

Figure 9: Ratio of actions changed across poisoned states in all four environments. Values are measured as the number of actions changed divided by the total number of timesteps the attack has poisoned.

### A.3.3 COMPARISON OF REWARD PERTURBATION MAGNITUDES

In this subsection we explore how the reward perturbation magnitudes of Q-Incept compare to those of SleeperNets and TrojDRL both with and without clipping in Figures 10 and 11. In both figures we can see just how large the unbounded reward perturbations of SleeperNets and TrojDRL are – inducing rewards as large as $\pm 55$ on Highway merge, which has natural rewards in the bound of $[0.25, 1]$. Under bounded reward constraints all attacks have reward perturbation levels that fall within a similar range. This indicates that adversarial inception is the key contributing factor to the success of Q-Incept.

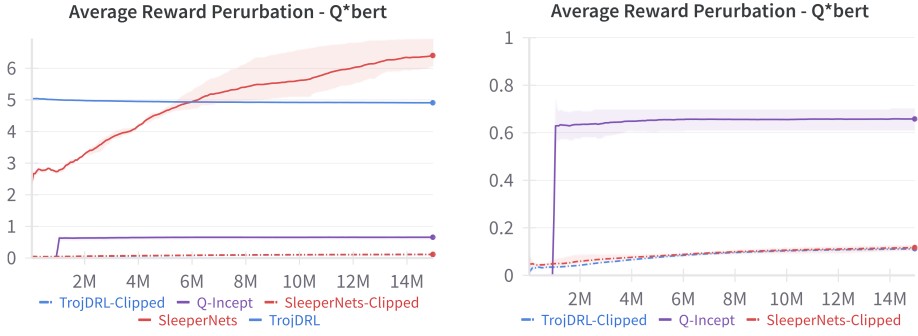

Figure 10: Comparison between the reward perturbations of Q-Incept against the baselines without clipping (left) and with clipping (right) on Q*bert.

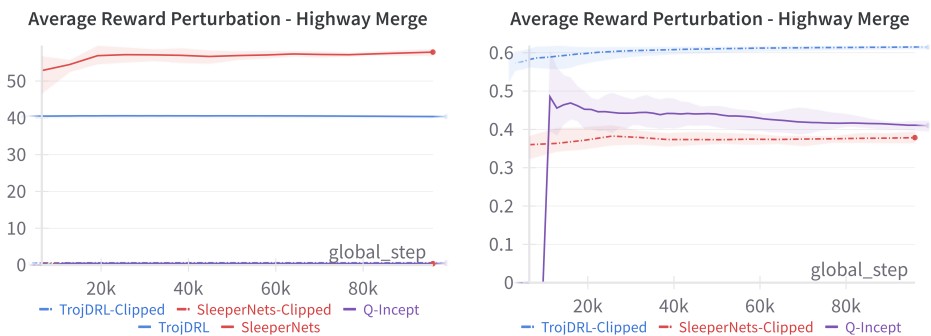

Figure 11: Comparison between the reward perturbations of Q-Incept against the baselines without clipping (left) and with clipping (right) on Merge.

### A.4 FURTHER DISCUSSION

In this section we provide further discussion on design choices made in this paper which were unable to fit in the main body.

### A.4.1 MOTIVATION FOR BASELINES

As mentioned in Section 6 we compare our Q-Incept attack against SleeperNets and TrojDRL as they represent the current state of the art for ubounded reward poisoning and forced action manipulation attacks respectively. For SleeperNets there are no other existing, ubounded reward poisoning attacks, so this decision is fairly clear. For TrojDRL there are other attacks which utilize static reward poisoning and forced action manipulation, however most only apply to specific application

domains like competitive, multi-agent RL (Wang et al., 2021) or partially-observable settings utilizing recurrent neural networks (Yang et al., 2019; Yu et al., 2022).

The only other, somewhat comparable attack is BadRL (Cui et al., 2023) which builds upon TrojDRL by optimizing the adversary's trigger pattern to achieve greater attack success. Trigger optimization is effective but orthogonal to the goals of this work as it can be generically applied to any attack. Furthermore, Rathbun et al. (2024) showed that BadRL without this trigger fine-tuning often performs worse than TrojDRL, likely since it uses methods to poison the most important – and thus hardest to poison – states in the MDP. Taking all of this into consideration we decided to omit BadRL from our empirical study. Therefore TrojDRL is the best baseline to use when comparing against static reward poisoning attacks using forced action manipulation.

### A.4.2 MOTIVATION FOR ENVIRONMENTS

In this paper we study 4 environments – Q*Bert, Frogger, Safety Car, CAGE-2, and Highway Merge. In the TrojDRL paper the authors focused their empirical studies towards Atari game tasks in the gym API Brockman et al. (2016). We think it is useful to include some of these environments like Q*Bert, as they are standard baselines for RL in general, however we believe it is critical to extend this study to further domains when studying the potential impacts of adversarial attacks. This belief is supported by the findings of Rathbun et al. (2024) who showed that TrojDRL – which consistently attains near 100% ASR on Atari environments without bounded reward poisoning constraints – often fails to achieve high ASR when tested on non-Atari environments.

Thus, to extend our study beyond the confines of Atari, we chose three other environments within the gymnasium API, allowing our code to work seamlessly between environments. We first chose Highway Merge since it seemed to be the most difficult environment for attacks to poison based upon the results of SleeperNets. Next we chose CAGE-2 as it not only represented a safety and security-critical domain, being an application of RL to cyber-network defense, but also because it uses non-image observations. Lastly we selected Safety Car, also from the environments studied in SleeperNets, as it represents a simulation of real-world, robotic applications of RL and, similar to CAGE-2, uses non-image observations.

