# OpenReview forum: "Adversarial Inception for Bounded Backdoor Poisoning in Deep Reinforcement Learning"
_ICLR.cc/2025/Conference — Submitted to ICLR 2025_

### Official Review · Reviewer_dgWL · 2024-10-28

**Soundness:** 2
**Presentation:** 2
**Contribution:** 2
**Rating:** 6
**Confidence:** 4

**Summary:**

The authors interrogate the drivers of current attack performance in RL, and in doing so identify a simple mitigation that affects multiple extant attacks, but, crucially, not their own novel attack framework.

**Strengths:**

The authors take an interesting perspective on RL attacks that highlights a weakness (vis a ve detectability) in current attack frameworks, that when resolved, opens new opportunities for highlighting more realistic attacks. This is then used to demonstrate the impact of their approach.

While there is a lack of clarity and specificity relating to the mathematical details of their work in the opening chapters, the remainder of the work is well written and cleanly presented. The authors have clearly thought about how to convey their intended message,

**Edit (2 December 2024)** - score has been increased to a 6 based upon the rebuttal period, however this is a soft 6. The authors have improved their work in terms of the rebuttal revisions, and have demonstrated a significant desire to improve their paper. However the rebuttal comments have also raised additional concerns regarding the clarity of explanations around core concepts (see discussion regarding the outer-loop threat model), and on if the authors have fairly captured relative performance.

The discussion period also did illuminate additional concerns regarding data handling practices that I am going to be charitable and assume have all been resolved. However for future reference I think that the authors need to fundamentally re-think how they're managing their data, as having multiple cut and paste related errors suggests serious process failings. Please generate data once, consistently, and then copy from that. Your problems have been created by your own piecemeal approach.

**Weaknesses:**

To begin with, I'm quite suspicious about the results in Table 5. Take for example Q-Incept, with a BRR of 99.29% and a SD of 17.27%. To maximise the SD assume that the first 3 trials were 100, and leave the last two trials as free variables. There's no solution in [0,100] that yields an average of 99.29% with a standard deviation of 17.27%. I don't see how some of the results are achievable for both CAGE-2 and Q-BERT - the standard deviations are just too high to be realisable.

Another major concern is the assumption that all attacks are evaluated under bounded reward poisoning (line 441 and 442). I appreciate that this is a simple and effective approach when demonstrated in Figure 2, but I am concerned about the lack of specificity to it, and the potential for this to be a highly environmentally sensitive approach (see Questions). The impact of this issue is amplified when Figure 4 demonstrates how intrinsic this approach is for the observed performance deltas.

I also find the argument about access to the machine and access to ram being a likely attack vector to be highly optimistic. These listed system intrusions are not contextualised by the number of systems that could be compromised, nor if they are at all related to any machine learning related systems. This is not to say I don't disagree about making white-box assumptions, but I would argue that there are far better arguments for it than system penetration statistics.

Speaking of arguments regarding the threat model, I also find the argument that offline trajectory manipulations are easier to detect as the relationship between states and actions would need to be well defined - in many systems of interest establishing such a model would either be a) trivial or b) potentially an important part of model development. In both cases, this would appear to undercut the argument being made.

Beyond these points, the authors show a consistent lack of specificity when it comes to their mathematics terms (see below), and while they mention testing at other beta ratios, but don't show them in the main body of the text.

Beyond these points, I also have a number of minor quibbles:
- Figure 1 attempts to demonstrate how the approach works. But it fails to detail how the agent would behave in the absence of a trigger? Moreover a reader may be confused as to why the agent chooses an action that is different to the transition action. In the context of trajectory manipulation it makes more sense, however this point is not well made in the opening of the paper.
- Table 1 should cite the other techniques - especially when this is the first time in the paper that they have been discussed.
- The authors perspective on the literature is heavily weighted towards reward poisoning attacks, but there's also more general poisoning attacks, or offline specific attacks. There's a broad topology of attacks against RL, including those that manipulate rewards, obversations, policies, and actions.
- S3 "The adversary attempts to instead train the agent on adversarial MDP" - is ambiguous and poorly phrased. For starters, it should be "on the adversarial MDP", as you're introducing a formal definition of it afterwards. But by discussing this as both an adversary (who is doing the action) and an adversarial MDP, there is an ambiguity about what the adversarial MDP is adversarial to. Moreover, this whole framing of "attempting to instead train" may be confusing to readers - how are they attempting to train this? Why is this only an attempt, is this not a guaranteed process? Does this MDP mechanicistically replate to M? Is it that M is replaced - if so, then a different phrasing would be more appropriate.
- "We model the trigger" - the trigger is not yet defined, so you're discussing a concept in your problem formulation for which the reader has no context to understand its meaning, nor how it relates to the sentences that have just been introduced. This definition on line 114 becomes extra confusing around line 125 - when $\delta$ is now the poisoned state, rather than being a trigger. It may well be that the trigger being activated converts $s$ to its poisoned state equivalent, but the current way this is presented does not make that point cleanly. This issue pops up later in S4.2 as well - and I guess it comes down to a differences in perspective. The authors are presenting the trigger as the system that is triggered to change between states, but I would think that the average reader would interpret a trigger as being something that induces a second system to change state behaviour.
- Repeated and superfluous definition of MDP's from line 87 and line 110.
- $\mathbb{S}$ is never introduced as (presumably) the set from which S can be drawn from until 6 lines after S is introduced, and 5 lines after $\mathbb{S}$.
- $\pi^{+}$ is not defined.
- Section 3 uses $S$, but Section 3.1 uses $s$ - seemingly for the same concept.
- Line 133 - V is not defined.
- $\mathbb{1}$ is not defined - although most would get what it is, it is still a point of notational complexity.
- Statistics presented as "in 2024 alone" for a paper that was written at some time in 2024 are impossible to parse by a reader, as they have no context to understand how far into the year these numbers were collected.
- Line 237 "are standard throughout [the] poisoning literature"
- When talking about $\beta$ - this is referred to as a rate/proportion, but it is only discussed in terms of states.  But is this the proportion of states across a single trajectory, or across all trajectories? There is no specificity to this point.
- Line 240 "decreasing the agent's performance in the benign task" - there's an argument to be made about why this is important, but instead it's left to the reader to understand the context here.
- L266 "they receives" -> receive.

**Questions:**

- The authors make a point about other approaches allowing unbounded perturbations. But part of this work is that stealthiness is a concern in prior works, and that multiple definitions exist. Do these alternate approaches allow unbounded perturbations /while/ maximising stealthiness? If so, does the fact that prior works allow unbounded perturbations have any practical impact?
- Figure 2 appears to be a very simple, but powerful argument against some of the tested comparisons. However, from my understanding, rewards for highway merge are highly sparse and spiky - so wouldn't these seem spikes be observed in the case of a weakly performing agent? If so, is it really that Figure 2 is showing anomalous spikes, rather than it just demonstrating the inherent spikiness in the rewards for this environment? Is this same behaviour seen with these attacks in other environments, or over longer time periods? How do these markers change between trained, untrained, and partially trained models?
- Could you explain the standard deviations in your table of results (see in the weaknesses for more details about this point of concern)?

---

> ### Author Response · Authors · 2024-11-22
> **Response to Reviewer dgWL**
>
> ## Summary
>
> We would like to first extend our genuine thanks to the reviewer for their thorough reading of our paper and the effort they put in, especially in the final section of their review. We have gone through all of the reviewer’s proposed changes and have improved the writing and presentation of our paper accordingly. Throughout this rebuttal we will refer to the updated and much improved manuscript which we have already uploaded. Highlighting a few of the improvements, we have:
>
> * Added an additional panel to figure 1 to include discussion about the agent’s behavior in benign states, and have updated the language in the caption to better reflect our attack methodology
> * Expanded our related work section to include discussion of other attack and defense types
> * Included discussion on other potential threat vectors compatible with our attack’s threat model including malicious training software, malicious cloud training, or attacks against offline RL
> * Moved additional experimental results over different beta values from the appendix to the main body in table 3
> * Improved our experimental setup discussion to resolve questions over unrealistic standard deviation values.
> * Made numerous improvements to our language and notation to disambiguate or resolve each of the reviewer’s “minor quibbles”
>
> In the remainder of the rebuttal we will go through each of the reviewer’s arguments and questions in order, providing in-depth responses with relevant citations and analysis. Upon reading our response we respectfully ask the reviewer to consider increasing their score. We otherwise invite the reviewer to bring up any additional questions or concerns they may have about our work.
>
> ## Addressing Weaknesses
>
> **1**. We understand your concern here and will provide a thorough explanation. The standard deviations are calculated over 5 seeds for each environment which are themselves averaged over 100 evaluation runs. The reason for the discrepancy you observe is that BRR can actually go above 100% on individual runs or even on average. Your proposed argument relies on the assumption that every sample is bounded at 100% which is not true. The actual, average episodic returns for each of our Q-Incept poisoned agents on CAGE are [-40.57, -41.59, -52.07, -37.29, -55.91] which has a mean of -45.49 and a standard deviation of 8.03. Unpoisoned agents receive a mean score of -45.17, thus our poisoned agents have a BRR of 99.29% and a standard deviation of 17.79%. Note here that the first, second, and 4th runs all have BRR scores above 100%. Again, we understand your confusion, and have added more explicit definitions of BRR and ASR in the paper. We have additionally mentioned in the caption of table 3 that BRR is clipped at 100% which was also done in the BadRL, TrojDRL, and SleeperNets papers.
>
>
> **2**. Evaluating attacks under bounded reward poisoning constraints is the main focus of study in this paper, as motivated in section 3.2 - without bounded reward poisoning constraints, all attacks can be trivially detected! Every benign reward function R, as specified by the benign MDP, has a natural upper and lower bound (otherwise RL methods won’t converge). If the victim knows these bounds then any rewards they observe outside the bounds are clearly suspicious and worthy of investigation. For instance, say you are training on Atari Q*bert with rewards clipped to [0,1], as is standard practice. If you then observe a reward of 15, you know something is wrong. This is what Equation (5) captures. Other details like the sparsity of the reward function, its general smoothness or behavior, etc. aren’t relevant to our analysis - only the bounds matter.
>
> Therefore evaluating each attack under these constraints is both fair and well justified. If we evaluated against unbounded SleeperNets the rewards would be so extreme that, under scrutiny of the detector in equation 4, the attack would be easily and immediately detected, making the results meaningless. Also see our response to question 1 for more details.
>
> **3**. We agree that it is important to further explore other potential threat vectors which may lead to similar levels of adversarial control as RAM access. As such we have expanded our threat model section to include references to malicious training software, malicious cloud training, or attacks against offline RL.

---

> > ### Author Response · Authors · 2024-11-22
> > **Response to Reviewer dgWL - Continued**
> >
> > **4**. You raise valid questions, but we believe such detection is actually computationally infeasible for the defender. Let’s first assume the defender has access to a simulator/model and hopes to detect adversarial inception during training or testing. The defender would first need to store the complete internal state of every timestep in each trajectory training. This is already highly memory and computationally expensive as internal environment states include magnitudes more information than the agent’s observations. After this the defender would need to simulate an additional environment step at each timestep given the agent’s, potentially adversarially manipulated, action to compare it against the observed outcome. This effectively requires double the computational cost of normal training as each time step needs to be simulated twice. For stochastic environments additional simulations may be required to reduce uncertainty, increasing training times significantly and leading to many false positives.
> >
> > In this paper we propose a new class of backdoor poisoning attacks against reinforcement learning, which motivates further research into the defense techniques you propose. Prior to this work, all attack methods relied on arbitrarily large reward perturbations to be successful, and defense techniques were designed around this assumed attack structure [2]. Our theoretical and empirical results strongly motivate further research into defense techniques against adversarial inception, along with more evasive attack strategies. We are excited to see what future works can achieve by building upon our strong foundations.
> >
> > **5**. In this final point we address all your comments listed after “Beyond these points…”. We have gone through each point individually and have improved and updated our paper accordingly. We have added our experimental results over different $\beta$ values to the main body of the paper, improved our first figure to include discussion about the agent’s behavior in benign states, added additional citations and discussion on other adversarial RL literature, and fixed all the notational and grammatical mistakes the reviewer has pointed out. The only point we did not directly address was in reference to Table 1 as it was moved to the appendix to save space given the additional experimental results requested by other reviewers. We also maintained the “repeated definition” of MDPs as the related work section defines the class of all MDPs while section 3 specifically defines the victim’s MDP. The notation is the same, but we want it to be clear that things like $R$ refers specifically to the victim’s benign reward function.

---

> > > ### Author Response · Authors · 2024-11-22
> > > **Response to Reviewer dgWL - Continued**
> > >
> > > ## Addressing Questions
> > >
> > > **1**. Some test time attacks measure stealth in terms of detectability for testing time attacks [3] but other backdoor attack works either define attack stealth informally[4] or use the definition from equation (2). We are the first work that considers bounded reward poisoning for backdoor attacks as an additional constraint, while all other attacks allow unbounded rewards. The practical implications are that unbounded attacks can be effective without adversarial inception, but are much easier to detect, as is discussed in section 3.2. On the other hand, bounded attacks require additional, novel approaches like adversarial inception in order to guarantee success. We believe the ease with which unbounded attacks are detected makes them less interesting to study. Thus, we hope future work will focus their efforts to better understand bounded attacks like adversarial inception.
> > >
> > > **2**. While merge does have “spiky” rewards, it also has tightly bounded rewards - they never go outside [0.25, 1]. This means the reward patterns displayed under TrojDRl and SleeperNets (reaching far beyond $\pm 5$) will never show up in the agent’s rewards unless they are being poisoned. This is the motivation for our reward poisoning constraints seen in Equation (5) - any reward observed outside the natural bounds of the benign reward function **must** be adversarial.
> > >
> > > These markers from TrojDRL and SleeperNets will be consistent no matter the point in training. TrojDRL’s reward poisoning is static (Eq 3) so the spikes will always be the same ($\pm c$), SleeperNets is dynamic (Eq 3) so the spikes vary slightly but will always extend outside the range [0.25, 1] for a sufficiently large alpha or c.
> > >
> > > In short, under our designed reward poisoning constraints the reward signal from a poorly or well trained agent can be easily distinguished from those induced by unbounded or static reward poisoning.
> > >
> > > **3**. Yes, we believe that this question was answered in our response above, but if you have any further questions or thoughts please feel free to let us know.
> > >
> > > ## Conclusion
> > >
> > > We again thank the reviewer for their feedback. We have addressed all the reviewer’s questions, improved the writing of our paper, and included additional experimental results. We again kindly ask the reviewer to consider increasing their score.
> > >
> > > [2] Chen et. al. “BIRD: Generalizable Backdoor Detection and Removal for Deep Reinforcement Learning”
> > >
> > > [3] Nasvytis et al., “Rethinking Out-of-Distribution Detection for Reinforcement Learning: Advancing Methods for Evaluation and Detection”
> > >
> > > [4] Yang et. al, “Design of intentional backdoors in sequential models”

---

> > > ### Comment · Reviewer_dgWL · 2024-11-24
> > >
> > > **Responding to your second rebuttal**
> > >
> > > I fundamentally disagree about the point about detecting state transitions that are poisoned in an offline model being computationally infeasible. You state "This is already highly memory and computationally expensive as internal environment states include magnitudes more information than the agent’s observations. After this the defender would need to simulate an additional environment step at each timestep given the agent’s, potentially adversarially manipulated, action to compare it against the observed outcome."
> > >
> > > I do not see how this would require the complete internal state of every timestep into memory at once. Why could the defender not validate the omdel by just looking at things on a timestep-by-timestep basis? And even if it did require "double the computational cost of normal training" - on the scale of published adversarial defences, doubling the computational cost at training is not outside the realms of possibility. In fact, I would argue that a lot of defences would be thrilled if they only doubled the computational cost.

---

> > ### Comment · Reviewer_dgWL · 2024-11-24
> >
> > Thank you for your response, and for clarifying the nature of BRR. Due to the nature of your response being over three messages, while I'm loathe to do this I'm going to split my response as well to ensure continuity.
> >
> > Unfortunately, this response has raised additional concerns for me. In particular, I'm concerned about the reliance upon the BRR metric as a tool for measuring performance. BRR is a metric that has only been used by your group (please don't break double blind, but there's only one group of authors using this metric, all of whom write on the same topic as this paper, so I am 100% confident that you are the same people), but I'm concerned that it intrinsically biases towards certain kinds of performance.
> >
> > Consider two different performance deltas - additive and multiplicative. If performance deltas are multiplicative, then BRR would capture that. However, if they are additive, then BRR is going to be intrinsically biased towards smaller values of V - due to the larger impact on performance changes. I don't think you've made the case that the impact of attacks being multiplicative, and, in the absence of that, I'm concerned about BRR as a metric.
> >
> > To clarify on the standard deviations, is it that if X is your signal, and SD is your standard deviation, is X a 1x5 vector (5 trials, each averaged over 100 samples), or is it a 1x500 (5 trials, 100 samples per trial)? Because otherwise even if BRR can go over 100, ASR is bounded in [0,1], and if X = [1x5] then things like the 1.5% column of table 4 would have incorrect values.
> >
> > As to your point that unbounded reward poisoning constraints, all attacks can be trivially detected - the point in my original review was to ask if stealthiness can be achieved either by a) norm minimising with unbonuded constraints on the norm (which, if the norm is minimised, then no, not all detects can be trivially detected) or b) by being unbounded in a norm that is not being considered by any detection mechanism. The point being is that there is some nuance about being unbounded.

---

> ### Author Response · Authors · 2024-11-25
> **Response to First Official Comment**
>
> We first kindly ask that you update your comment to remove all discussion about who you think we are as this has no bearing on the paper, you may very well be incorrect, and **you may bias other reviewers**.
>
> BRR is not a metric exclusive to SleeperNets in the prior work, it was also used in BadRL [5] under a different name “Clean Data Accuracy” or CDA. The motivation for using BRR is its immediate interpretability and comparability across environments of differing return scales. If the reader sees a (delta) score of -0.31 on Merge versus -295.5 on Q*Bert it’s not obvious that these are both very small relative to the benign returns of 15.16 and 17100 on each environment. With BRR this is more immediately and neatly captured, resulting in scores of 97.97% and 98.49% respectively. If the reviewer has an additional metric they wish us to use we can certainly consider including it in the main body. Perhaps something like table 3 in BAFFLE [6]?
>
> Currently there is no way for us to know if the deltas induced by poisoning are additive or multiplicative, however we believe it does not make a significant difference. Across environments and attacks we see that the difference between benign and poisoned agent performance is very minimal, with the only real exception being Safety Car. This is further verified by the training curves we give in the main body and appendix, which are indistinguishable between poisoned and benign.
>
> On your point of standard deviations, they are calculated over a 1x5 vector rather than 1x500 since this captures variance across *training runs* rather than episodes. We also appreciate you pointing out the mistakes in table 4. Amidst the rush of the rebuttal we accidentally submitted a PDF with out of date values in table 4 (see, for instance, that most of the 1.5\% column was copied from the 1.0\% column as temporary values). This is further proven as we referred to a value of 87\% in the text which did not appear in the incorrect table. We have since updated the document to include the correct values, which don’t deviate significantly from the originals. We have gone through all the data in our paper again and were unable to find other similar mistakes. The raw ASR data for the 1.0\% column of table 4 (which you were effectively referring to) is as follows:
>
> * Q-Incept: 99.90%	66.14%	100.00%	100.00%	100.00%
>
> * SleeperNets: 0.26%	3.38E-05%	2.31E-05%	1.88E-05%	8.36E-05%
>
> * TrojDRL: 5.66E-07%	8.62E-08%	0.145231694%	1.20E-05%	0.1367142648%
>
> For our new 1.5% data the ASR values are:
>
> * Q-Incept: 99.93054867%	99.9998033%	100%	100%	100%
> * SleeperNets: 0.06%	2.94E-05%	1.57%	0.51%	0.61%
> * TrojDRL: 1.16E-05%	3.13E-06%	6.64E-06%	0.08633258939%	0.0256%
>
> Which results in the ASR and standard deviation scores we claimed within rounding to the hundredths place. If there is any further confusion on the validity of our data, which we have verified twice, please let us know.
>
> There are certainly additional bounds one could consider, such as $l_2$ over all poisoned rewards, but they must adhere to Equation 5 to avoid detection by Equation 4. Other backdoor attack works do not consider such bounds to our knowledge. When we refer to bounded rewards we are exclusively speaking with respect to equation 5.
>
> [5] Cui et. al. “BadRL: Sparse Targeted Backdoor Attack Against Reinforcement Learning”
>
> [6] Gong et. al. “BAFFLE: Hiding Backdoors in Offline Reinforcement Learning Datasets”

---

> ### Author Response · Authors · 2024-11-25
> **Response to Second Official Comment**
>
> > "I do not see how this would require the complete internal state of every timestep into memory at once."
>
> Thank you for the reply, your questions here are valid but missing some key details of our threat model. In section 3.3, as well as algorithm 1 we define our attack as using the “outer loop” threat model where the adversary manipulates the agent’s experience data *after* each episode, not during. A defender who performs their analysis on a per-timestep basis as you describe would be analyzing exclusively *clean, unpoisoned* interactions with the environment. Therefore, the defender would need to store all the state information for each time step to detect Q-Incept after each episode. This, in addition to the computational cost of additional simulations and the uncertainty of stochastic environments, is why we claim the defense is currently computationally infeasible. We agree that it is certainly possible for a defender to devise a strategy to detect our attack, but it would likely be non-trivial - requiring additional follow-up work to fully develop and analyze.

---

> ### Comment · Reviewer_dgWL · 2024-12-01
>
> Just to clarify - I assume that these updates values are out by 2 sig figures on the first two values of Q-Incept? Ie. that thy're 99.93% and and 99.99%? Because otherwise something is far more wrong with your data than had appeared before. Polarised values of essentially 1% or 100% speak to far broader issues in your experimental set up than had initially been apparent. Do you have any explanation as to why they are so polarised?
>
> As to the outer-loop threat model - a problem here is that you don't define this. If it is a core and crucial part to understanding your attack process and threat model, then having one sentence that references it - and a sentence that does not even cite the original source - is patently not sufficient. So I would suggest that if I'm missing a key detail of your threat model, then it may be because that content could have been more clearly defined and explained.

---

> > ### Author Response · Authors · 2024-12-02
> > **Response to Reviewer dgWL**
> >
> > ## Response
> >
> > Yes, thank you for pointing out the mistake. We copied the data over from our spreadsheet and didn't notice the discrepancy between how the data was represented (as a [0,1] ratio vs [0,100] percentage), we have updated our comment to fix the mistake. Our aim in writing this paper is to better understand a critical threat against reinforcement learning systems. We are not here to deceive reviewers with data tricks.
> >
> > For the outer loop threat model we think our current explanation is far from "patently not sufficient". We discuss:
> > * what the adversary can observe (training trajectories/data)
> > * what they can manipulate (states, actions, rewards)
> > * and when they can manipulate it (after each episode but before policy optimization).
> >
> > The threat model section is fairly short, but important, so we expect readers to be able to read through and understand its contents.
> >
> > Furthermore, the threat model is baked into Algorithm 1 - we directly, algorithmically show how the adversary sees the training trajectory, selects and manipulates a subset of it, and then the victim optimizes with respect to the poisoned data. There are then further references to this in the text e.g. "... incepting false values into the **replay memory** $D$...".
> >
> > If the reviewer still thinks this amount of context and explanation is insufficient then we can certainly add a few more sentences to the threat model section or reword some parts. We will also add another, proper citation next to our textual citation of SleeperNets in the first sentence of the threat model section.
> >
> > ## Summary of Dialogue with Reviewer dqWL
> >
> > We have had an extensive back and forth with the reviewer, which has resulted in many, significant improvements to our paper which have been greatly appreciated. We now kindly ask the reviewer to reflect upon the larger context of this dialogue. First, we have improved our paper by:
> > * Adding new experiments over different beta values on *all* environments.
> > * Including new experiments on the Frogger environment.
> > * Performing an ablation over the poisoning rate on CAGE-4.
> > * Significantly improving the technical and mathematical writing of our paper, leveraging nearly *all* of the reviewer’s suggested changes.
> > * Adding more details to our figures, threat model, and related works sections per the reviewer’s request.
> >
> > In addition to this we have rebutted all of the reviewer's complaints about our paper including:
> > * Questions over the detectability of our attack - explaining the difficulty of training time defenses against adversarial inception
> > * Concerns over the validity of our data - providing proof of our experimental results
> > * Concerns with our choices of metric, and our choice of bounded reward poisoning - providing thorough explanations justifying our decisions
> >
> > ### Conclusion
> >
> > This review process has been lengthy, and we believe that we’ve lost sight of the contents of the paper itself. Therefore we want to reassert our numerous contributions. In this paper we:
> > * Display the triviality of detecting reward-dependent poisoning strategies like SleeperNets and TrojDRL
> > * Explore how these attacks lose all guarantees and almost always fail when forced to clip their rewards, as shown in our empirical results
> > * Propose and develop a new class of backdoor poisoning attacks which:
> >     * are the first backdoor attacks to leverage theoretically motivated action manipulation strategies
> >     * are the **first and only** attacks which can **theoretically guarantee** attack success while minimally perturbing the agent’s reward signal
> >     * require the same level of adversarial access as the prior, less effective attacks
> > * Lastly, we provide extensive, empirical results showing that our proposed attack significantly outperforms the baselines - achieving 100% ASR on almost all environments while SleeperNets and TrojDRL can fail to achieve above 6% ASR.
> >
> > We strongly believe these contributions are significant to both the broader Reinforcement Learning and Adversarial Machine Learning communities - particularly our theoretical contributions in Section 4, our technical contributions in Sections 3 and 5, and our empirical contributions in section 6. We kindly ask the reviewer to again reflect upon these contributions and consider increasing their evaluation of and confidence in our paper.
> >
> > If the reviewer has any further questions we will gladly continue the dialogue.

---

> > > ### Comment · Reviewer_dgWL · 2024-12-02
> > >
> > > To be clear - my comment about the outer-loop specifically relates to lines 192-204. Perhaps describing it as "patently not sufficient" was gilding the lily slightly, but I do think that if the outer-loop framing is so crucial to understanding your process, then it should be explicitly referenced. Using "this model" when discussing the threat model may be sufficient to tie the content to the outer-loop invites ambiguity, which is not helped by the outer-loop threat model only being referred to once in the entire paper - yet, as you've described it in the rebuttals, it's crucial to understanding your process. Also, again, it also was not cited properly.
> > >
> > > As to the rest of this - I appreciate your perspective, and that the adversarial nature of the review process is stressful. However I will be making up my own mind regarding where my score sits. You can state that you've resolved issues with, say, the metrics, however I personally have to interpret statements like "Currently there is no way for us to know if the deltas induced by poisoning are additive or multiplicative, however we believe it does not make a significant difference" as the type of things that one would say if they wanted them to be true. Fundamentally, as a reader I have no way to know how the distribution of values affects your results. I also have to factor in to my analysis that you have, as you have acknowledged, made multiple data handling errors. This is another point of concern, because I can't validate all your work, all I can do is review based upon the concerns that I hold.
> > >
> > > On the flip side of course is that you have been very engaged over this period (which is a credit to you, although the amount of emails due to split, character limit evading responses did at times do my head in), and you have been willing to address criticisms and concerns. I'll have a think.

---

> ### Author Response · Authors · 2024-12-02
> **Response by Authors**
>
> Thank you for your continued engagement with us. We certainly want your review decision to be a reflection of your personal opinion. We just kindly ask you consider all pieces of our paper's contributions.
>
> In regards to the data handling errors, we apologize. This rebuttal process has been in the middle of a very hectic semester for us, so some mistakes were made. The final results present in the paper *are* correct, however, as we have double checked all of them.
>
> In regards to the deltas being multiplicative or additive, we genuinely don't know if there is a way to determine this. There are three, complicated systems interacting at once here: neural network gradient optimization, reinforcement learning environments, and backdoor poisoning. It is unclear if this interaction should result in deltas that are additive, multiplicative, or even something else w.r.t. our backdoor poisoning.
>
> By our theoretical results there should be no difference between the performance of the unpoisoned and poisoned agent, but Q-Incept is not a perfect replication of our theory so it's hard to say what does and what doesn't transfer.
>
> If you recall Q-Incept's poisoning strategy you'll note that the agent's monte-carlo estimate of their value in unpoisoned states is effectively unchanged after poisoning. We believe that discrepancies come from two, machine learning centric avenues: amount of training data, and generalization across similar states. Under poisoning the agent effectively trains on less clean data, which may result in slightly lower scores. Additionally, the agent's policy must generalize in some way across states given their neural network architecture, which may lead to some conflict between benign and poisoned states. We think generalization is the main reason for BRR scores on Safety Car being much lower as the agent's inputs are much smaller than other environments.
>
> Given all of this, it is unclear to us if the deltas should be multiplicative or additive, therefore we have use BRR as our metric of choice due to its interpretability for readers across environments. We have no particular attachment to using BRR as our attack stealth metric, and certainly aren't intending to use it for deception.
>
> As a show of good faith we will include tables with raw scores and additive deltas in the following tables. The first is on Q*Bert with $\beta = 0.3%$
> |   Metric   | No Poisoning |  Q-Incept  | SleeperNets |   TrojDRL  |
> |:----------:|:------------:|:----------:|:-----------:|:----------:|
> | Raw Return |    17,101    |   18,381   |    16,840   |   17,617   |
> |   Raw Std  |     1,169    |     882    |    1,982    |     909    |
> |    Delta   |      N/A     | **+1,280** |     -261    |    +516    |
> |     BRR    |      N/A     |  **100%**  |    98.48%   |    **100%**    |
>
> The next is on Safety Car with $\beta = 0.1%$
>
> |   Metric   | No Poisoning | Q-Incept | SleeperNets | TrojDRL |
> |:----------:|:------------:|:--------:|:-----------:|:-------:|
> | Raw Return |     11.4     |    9.17    |     9.22    |    10.86   |
> |   Raw Std  |     1.26     |    0.038   |     1.98    |    1.13    |
> |    Delta   |      N/A     |    -2.23   |    -2.18    |  **-0.54** |
> |     BRR    |      N/A     |   80.48%   |    80.94%   | **95.28%** |
>
> If the reviewer would prefer that our paper include additive deltas like this in our paper than we can certainly comply. Again, we have no attachment towards using BRR. In fact, in these cases BRR actually makes our results look weaker due to the 100% clipping we discussed earlier.
>
> Thank you again for your continued dialogue with us. Please let us know if there are any further questions.

---

> > ### Comment · Reviewer_dgWL · 2024-12-02
> >
> > Given it's the end of the review period, I'll state explicitly the idea that I've been trying to subtly suggest to you: looking at one averaged metric of performance is not enough. You need to look at the raw numbers, and the underlying distributions. That you don't know if the performance delta is additive or multiplicative is the fundamental problem here, and suggests you need to think about how to measure and assess that.
> >
> > Whenever you use an averaged metric there's always the risk that the average is being distorted by one part of the data range. How do you show that your performance is consistent? One way is to plot it out. Plot relative performance in terms of sorted size, for example. Look at papers from the field of Certified Robustness for example - every paper will have average metrics, but they'll also have performance vs size of output graphs, to capture if performance is a product of small changes or larger changes.
> >
> > I was at a talk at a conference recently by some of the core authors in the space of Instance Space Analysis (the MATILDA team out of Australia). Their whole thing is about how can we test models (including deep learning models) to make sure we're making fair assessments of their performance. While I'm not suggesting you should be using ISA, I do think that this work would benefit from really thinking about how best to fairly capture performance.
> >
> > Score has been raised to a 6, but for the AC's benefit this is a soft 6.

---

> > > ### Author Response · Authors · 2024-12-02
> > > **One Final Question**
> > >
> > > Thank you for increasing your confidence in our paper, we greatly appreciate it :). The reference to ISA is also really useful so we'll look more into that to see how those methods can be applied to our work. Finding the right metrics in this sub-field is certainly important.
> > >
> > > We think we understand the core of what you're saying, but we have some questions to understand exactly what you mean in the second paragraph. In our case what would the x-axis "sorted size" be? In works like "Certified Robustness of Nearest Neighbors against Data Poisoning and Backdoor Attacks" they plot certified bounds vs poisoning size (see figures 2-7), which in our case would be a function of $\beta$. Is this what you are referring to or is it something else?

---

> > > > ### Comment · Reviewer_dgWL · 2024-12-02
> > > >
> > > > To be clear - ISA likely will not be the solution to your problems. But it may be a source of inspiration for thinking about data quality and analysis quality.
> > > >
> > > > As to the second paragraph - treat it as inspiration, not a prescription. But you could plot, for example, mean(E[V_pi+] where E[V_pi] > c) for varying c. Or E[V_pi] < c. These are just suggestions to try and visualise where in your distribution is driving performance deltas.

---

> ### Author Response · Authors · 2024-12-02
> **One Final Question - Continued**
>
> Interesting, thank you for these suggestions. We hope you don't mind our further questions towards you on this topic, but we'd like to try and replicate the plots you're suggesting. Let's look at our raw data for Q*Bert, with both rows independently sorted by value
>
> |   Sample #   |     1    |     2    |     3    |     4    |     5    |
> |:------------:|:--------:|:--------:|:--------:|:--------:|:--------:|
> | No Poisoning | 16555.76 | 16796.71 | 17271.91 | 17910.39 | 18476.24 |
> |   Q-Incept   |   17709.01  |  17736.54 |  17838.52 |   18983.27  | 19641.25 |
>
> Would we then create our plot from the following table?
>
> | c            | 16555.76 | 16676.24 | 16874.79 | 17133.69 | 17402.20 |
> |:------------:|:--------:|:--------:|:--------:|:--------:|:--------:|
> | Q-Incept     |   17709.1  |  17722.77 | 17761.36 |   18066.84 |  18381.72 |
>
> Here the nth c row is the average of the sample 1-n rows in the raw data table, the Q-incept row is similarly averaged but for the Q-incept data. Sorry if we're misunderstanding, but it can be hard to interpret details with text alone. Thanks for following us through this,  we greatly appreciate your time.

---

### Official Review · Reviewer_L6Gf · 2024-11-01

**Soundness:** 3
**Presentation:** 3
**Contribution:** 3
**Rating:** 6
**Confidence:** 4

**Summary:**

This paper proposes a new set of backdoor attacks that minimize the altering of the target agent's reward function. The paper demonstrates the effectiveness of the proposed method both empirically and theoretically.

**Strengths:**

+ The paper proposes a new attack against DRL under the constraint of reward mutations.

+ The paper provides theoretical justifications for the proposed attack.

**Weaknesses:**

- I would appreciate the authors showcase or discuss the generalizability and scalability of the proposed attacks. The authors can consider reevaluating their attacks on more complex DRL environments, such as MuJoCo or Atari games.

- I would suggest the authors evaluate the proposed method against existing backdoor detection and defenses, such as [1].


[1] BIRD: Generalizable Backdoor Detection and Removal for Deep Reinforcement Learning

**Questions:**

1. How well the proposed attack can be generalized to more complex DRL environments?

2. Is the proposed attack applicable to multi-agent RL environments?

3. Is the proposed attack resilient to existing backdoor defenses for DRL?

---

> ### Author Response · Authors · 2024-11-22
> **Response to Reviewer L6Gf**
>
> ## Summary
>
> We thank the reviewer for their positive assessment of our paper and appreciation of our novel theoretical results. Throughout this rebuttal we will refer to the updated and much improved manuscript which we have already uploaded. The reviewer’s first concern was with the scalability of our method. In response to this we have conducted multiple additional experiments, including
>
> * full experimental results on the Atari Frogger environment - which strongly verify the scalability and versatility of our method
> * ablations over the poisoning rate $\beta$ on the CAGE environment - verifying the stability of Q-Incept with respect to the poisoning rate.
>
> The reviewer was also concerned with the ability of our attack to evade current defense techniques. In our rebuttal we provide an in-depth response to this concern - highlighting the incompatibility of [1] with Q-Incept without major modifications to the underlying defense method and codebase. Upon reading our thorough response we kindly ask the reviewer to consider increasing their score. We otherwise invite the reviewer to bring up any additional questions or concerns they may have about our work.
>
> ## Addressing Weaknesses
>
> **1**. We are highly confident in the scalability of our approach due to the strength of our theoretical results and their generalization across all MDPs. To further verify this we have added additional experimental results to the main body of the paper including evaluations on Atari Frogger, ablations over $\beta$ on CAGE, and additional results over smaller $\beta$ values on every environment in Table 3 (these results were initially in the appendix). All these new results add further strength to our claims of the scalability and effectiveness of Q-Incept across a diverse range of environments.
>
> **2**. We agree that additional discussion in regards to defenses is necessary and have thus cited this work in our related work and conclusion/discussion sections. We downloaded and modified the BIRD code to integrate with ours, and were able to successfully run it against our agents poisoned by Q-Incept. The results showed that BIRD was unsuccessful in reconstructing our attack’s trigger, however we have multiple reasons to not trust these results. Unfortunately the BIRD defense codebase was incompatible with our framework without significant modifications being made. The code of BIRD is heavily designed around RNN architectures being used in the agent’s models while our code uses CNN architectures and framestacks for the agent’s observations. Due to the short timespan for this rebuttal, along with the multitude of other experiments we ran for other reviewers, we do not have enough time to verify that our code changes did not disrupt the defense’s core method.
>
> In spite of this we do have strong reasons to believe the BIRD defense may not work well against our attack. The defense relies on the assumption that “the attacker needs to manipulate the victim agent’s reward function, assigning the agent an ultra-high poisoned reward when it takes the poisoned action at poisoned states“, however this assumption does not hold with Q-Incept as our reward perturbations are bounded and often small (see appendix A.3.3). The defense then uses the agent’s value function to see in which state perturbations cause this high, adversarial return in contrast to the, relatively smaller, expected return in the benign state. Under the construction of adversarial inception the difference between the agent’s expected value in poisoned states versus benign states is bounded by [L, U], assuming the final policy is optimal. In the case of Atari environments this bound is [0,1], meaning the difference in the agent’s expected value when seeing the trigger will not differ significantly from its expected value in benign states. This violates the underlying assumptions of the BIRD defense, leading us to believe it will be ineffective against Q-Incept.

---

> > ### Author Response · Authors · 2024-11-22
> > **Response to Reviewer L6Gf - Continued**
> >
> > ## Addressing Questions
> >
> > **1**. Based upon our additional experimental results on Atari Frogger we believe the attack scales very well to more complex environments. In fact, the attack’s performance seems to be strongest against these environments over the (relatively) simpler Highway Merge and CAGE tasks.
> >
> > **2**. Yes, the core methodology of adversarial inception can certainly be applied to multi-agent RL (MARL) and this would certainly be an important and interesting direction for future research. However, one would need to better define the backdoor adversarial objectives within the context of MARL: do we want to backdoor all the agents, do we just want to backdoor one of the agents, do we want to use the backdoor on one agent to exploit other agents in the system, is the MARL system centralized or decentralized? All these questions are very interesting but non-trivial. As such, we are excited to see the growth of this research over time, and are confident that adversarial inception can be used as a theoretically rigorous foundation to build upon.
> >
> > **3**. See our response to weakness 2 above.
> >
> > ## Conclusion
> >
> > We again thank the reviewer for their positive review of our paper. We have addressed all the reviewer’s questions, improved the writing of our paper, and included additional experimental results. We again kindly ask the reviewer to consider increasing their score.

---

### Official Review · Reviewer_q6gp · 2024-11-03

**Soundness:** 2
**Presentation:** 2
**Contribution:** 2
**Rating:** 3
**Confidence:** 4

**Summary:**

This paper introduces "inception attacks," a novel approach to DRL backdoor poisoning that overcomes the limitations of reward-based attacks by manipulating the relationship between chosen and executed actions during training, rather than relying on suspicious reward modifications. The method is formalized through the adversarial inception attack framework and implemented via "Q-Incept," which demonstrates superior attack success rates under bounded reward constraints. The effectiveness is theoretically proven and empirically validated while maintaining the agent's normal performance on its intended task.

**Strengths:**

1. Identifies a fundamental limitation in reward-based backdoor attacks and proposes a novel solution through action manipulation
2. Provides theoretical analysis for adversarial inception, with results generalizable across MDPs,
3. Proposes formal guarantees for maintaining dual objectives - normal task performance and backdoor effectiveness

**Weaknesses:**

1. The core concept of action manipulation between chosen and executed actions isn't novel, being previously explored in adversarial RL literature ([1], [2])
While the application to backdoor attacks is new, the paper fails to properly acknowledge and discuss these related works.
2. The paper's description of experimental scenarios appears overclaimed - **should not characterize simple gym environments as "robotics navigation"**. While the abstract and introduction claim diverse experiments, the evaluations are actually conducted in very simple Gym environments with minimal difficulty, raising serious concerns about empirical effectiveness and practical scalability.
3. The ablation studies on attack budgets are notably incomplete, lacking comprehensive exploration of different parameter settings and their impacts.

[1] Action Robust Reinforcement Learning and Applications in Continuous Control, Tessler et al, ICML 2019

[2] Game-Theoretic Robust Reinforcement Learning Handles Temporally-Coupled Perturbations, Liang et al, ICLR 2024

**Questions:**

See weaknesses

---

> ### Author Response · Authors · 2024-11-22
> **Response to Reviewer q6gp**
>
> ## Summary
>
> We would like to thank the reviewer for their insight since it has helped us greatly improve the empirical rigor of our paper. Throughout this rebuttal we will refer to the updated and much improved manuscript which we have already uploaded. One of the reviewer’s main concerns was with the breadth of our experimental results. In response to this we have conducted multiple additional experiments, including
>
> * Full experimental results on the Atari Frogger environment - which strongly verify the scalability and versatility of our method
> * Ablations over the poisoning rate $\beta$ on the CAGE environment - verifying the stability of Q-Incept with respect to the poisoning rate.
>
> We have additionally provided in-depth response and analysis in regards to the reviewer’s claims of insufficient novelty, noting the orthogonality of their cited works in terms of: method, domain, adversarial objectives, and threat model. We reassert that *test time attacks are not comparable to training time attacks*.
>
> Upon reading our rebuttal we humbly ask the reviewer to consider increasing their score. We otherwise invite the reviewer to bring up any additional questions or concerns they may have about our work.
>
> ## Addressing Weaknesses
>
> **1**. We appreciate the reviewer for highlighting these works and have cited them in the related work to add additional context to our contributions. We strongly disagree with the reviewer’s implication that the existence of these works diminishes the novelty of our methodology, however. These papers study **test time attacks** under the assumption the adversary has **direct access to the agent’s actions after deployment**. They further focus on attacks in which the agent is forced to take *sub-optimal* actions during *testing*, while our work theoretically proves that *optimal* actions during *training* can be leveraged to induce adversarial behavior during testing. This result is, in of itself, highly novel and worthy of publication. Furthermore, our proposed Q-Incept attack performs no forced action manipulation - instead altering the agent’s experience data such that they “think” they took the target action when they truly took and transitioned with respect to an optimal action. This is the exact opposite methodology of [1] and [2]. The two works are completely incomparable to ours and do not impact the novelty of our methodology or rigorous theoretical results.
>
> **2**. We appreciate this feedback, and have accounted for it by including additional experiments on the “Frogger” Atari environment in the updated manuscript. These experiments further reinforce that Q-Incept not only scales to more complicated environments, but is also the only currently developed attack capable of achieving high levels of attack success under reward poisoning constraints - attaining 100% ASR at a poisoning rate of 0.3% while the baselines fall below 50% ASR. We additionally updated our language in the introduction to be more fitting for the environments we used.
>
> We would also like to note that both our empirical methodology and theoretical results are built entirely on top of the well established foundations of DRL methods and RL theory. From our theoretical results in Section 4 we know that our core methodology of adversarial inception scales and generalizes to all MDPs. Our proposed attack, Q-Incept, then replicates this versatile framework and further leverages the capabilities and scalability of DQN. If DQN scales to more complex environments, so will Q-Incept. By contrast, the prior attacks don’t even perform well in “simpler” environments.
>
> Our motivation for including these “simple” environments is that the domain seems to matter most when analyzing backdoor attacks than the scale of the environment. Prior works like TrojDRL and BadRL only evaluated their attacks against Atari environments. While these are more difficult RL problems than, for instance, Car Racing, they greatly limit the domain scope of evaluation. As a result, in the SleeperNets paper they showed that these (initially claimed as) “universal” attacks of TrojDRL and BadRL, which achieved near 100% ASR on Atari environments, suddenly failed to achieve above 60% ASR when evaluated on environments from other domains like Highway Merge.
>
> This is the reason we have extended our study to multiple environments spanning different application domains, even if they seem simpler as RL tasks. To evaluate on “complex” environments like Atari alone would be insufficient. You can also refer to appendix section A.4.2 (included before the rebuttal) for further discussion on this topic.

---

> > ### Author Response · Authors · 2024-11-22
> > **Response to Reviewer q6gp - Continued**
> >
> > **3**. To address this concern we have added an additional ablation over the poisoning rate $\beta$ on the CAGE environment to the main body of the paper. This study shows that Q-Incept is the only attack which improves in terms of ASR as the poisoning rate increases. It achieves this while also maintaining a steady BRR score, displaying the stability of the attack’s stealth as the poisoning rate increases.
> >
> > Furthermore, we have moved additional experimental results at lower $\beta$ values from the appendix into the main body of the paper in Table 3. This provides a much larger breadth of experimental analysis to our paper while maintaining the strength of our results.
> >
> > ## Conclusion
> >
> > We again thank the reviewer for taking the time to read our rebuttal. We have provided additional experimental results to prove the scalability of our method and have refuted their claims of insufficient novelty. In light of this, we again kindly ask the reviewer to consider increasing their score.

---

> > > ### Comment · Reviewer_q6gp · 2024-11-27
> > > **Response to authors**
> > >
> > > Thank you for your response. I have several substantial concerns regarding your paper and responses:
> > >
> > > * Providing a thorough acknowledgment and discussion of related works should be considered a fundamental requirement for an ICLR submission. As a reviewer, I raised this point specifically to encourage you to clarify the distinctions between your paper and recent related works in this field.
> > > * I strongly disagree with your claim that "the domain seems to matter most when analyzing backdoor attacks than the scale of the environment." What is the basis for this assertion? Conducting comparisons without including the benchmarks discussed in related works is methodologically questionable.
> > >
> > > While incorporating additional domains is valuable, this does not justify making broad claims about these environments. To be specific, I must emphasize that Safety Gymnasium (https://github.com/PKU-Alignment/safety-gymnasium) cannot be considered representative of the robotics domain as it is not a mainstream robotics benchmark. The reason for requesting Atari comparisons is that it represents the most commonly used environment for RL attack research and provides the most straightforward way to compare against existing baselines.
> > >
> > > Given these concerns, I maintain my original assessment that this paper is not yet prepared to meet ICLR standards.

---

> > > > ### Author Response · Authors · 2024-11-27
> > > > **Response to Reviewer**
> > > >
> > > > Thank you for writing this follow-up comment. As authors our goal in writing the related work section is to properly acknowledge prior works and discuss their methods in relation to ours to give our work proper context. Unfortunately the space of backdoor poisoning attacks in RL is very small at the moment, so we have cited all related works to the best of our knowledge. We originally did not discuss [1] and [2] in our related work as they seemed to study very different problems from ours. We now see their relevance, and have greatly extended our related work section to discuss these and other related papers. Please let us know if there are other works you think we should cite.
> > > >
> > > > Our reason for the belief that "domain matters more than scale in back door attacks" is exactly as we described in our rebuttal - many attacks like TrojDRL or BadRL may work well in one domain (Atari) but can fail in other domains, even if the task is simpler. We also disagree that we have not "[conducted] comparisons without including the benchmarks discussed in related works". The current state of the art attack is SleeperNets, as such we have compared against 4 of its 6 baselines (including Safety Car). In addition to this we have compared against 2 of the baseline environments from TrojDRL. Across all these shared baselines we greatly outperform both attacks.
> > > >
> > > > Our reason for studying Safety Car over other robotics environments is twofold - it was a baseline from SleeperNets, and other robotics environments (like MuJoCo) use continuous action spaces which are not compatible with our attack formulation. It is certainly possible for our attack to be extended to continuous action spaces, but this would require additional, novel methodology which has yet to be developed in the literature. Safety Car represents a robotics adjacent environment where we can properly discretize actions without harming performance.
> > > >
> > > > RL is a tool which is being applied to a multitude of domains. Our goal as security researchers in RL is to study how adversarial attacks can impact each of these respective domains. As such we have chosen environments from a wide range of representative tasks to show the transferability of backdoor attacks across domains. Unfortunately we cannot research the state of the art in every sub-field of RL, but we believe that the breadth of empirical results sufficiently motivate further research into defending against adversarial inception attacks.

---

### Official Review · Reviewer_pHDQ · 2024-11-08

**Soundness:** 2
**Presentation:** 2
**Contribution:** 2
**Rating:** 5
**Confidence:** 4

**Summary:**

The paper studies the problem of backdoor attack in Deep Reinforcement Learning termed as “Adversarial Inception”. Unlike prior works that require large perturbation in reward to achieve attack goal, this paper proposes a novel attack method in bounded reward poisoning setting but allowing the attacker to also perturb the actions taken by the agent(hence the name inception). The authors propose a training algorithm called Q-incept to install the backdoor behavior in the agent which can be triggered during deployment using predefined trigger functions. The paper provides some theoretical results on their attack algorithm and supplement it with various empirical results to prove the efficacy of their method on various simulation environment.

**Strengths:**

1. The paper proposes novel attack strategy in bounded reward domain that require changing the action taken by the agents.

2. The paper appear to provide both theoretical to guarantee effectiveness of their training algorithm.

3. The paper  conducts empirical analysis comparing their algorithm with prior works like SleeperNets, TrojDRL to illustrate superiority of their method in bounded reward setting.

**Weaknesses:**

1. The problem setup is poorly formulated. There are a bunch of issues with notation and objectives defined in the paper. See the question section for more info.

2. Threat model is pretty strong and makes it less practical in real world setting - it requires the adversary to break into the training system and access RAM values and change all of state, actions and rewards. Specifically, unconstrained inception attack is very strong. If the adversary is that powerful, why wouldn’t it directly modify the final policy itself?

3. The action based attack are not new and has been previously studied more rigorous RL attack setting. See [1].

4. Experimental results are very poorly states. The authors have just stated a plethora of results without highlighting any interesting takeaway points. Metrics like Attack Success Rate and Benign Return Ratio are not defined anywhere in the paper.

5. The paper does not discuss sufficient details on defenses against their attack method.


[1]. [Optimal Attack and Defense for Reinforcement Learning](https://ojs.aaai.org/index.php/AAAI/article/view/29346)

**Questions:**

1. Why is a good policy a map from superset of states of actions? How does actions not in $M$ affect the transitions in $M$?

2. What does maximizing over $M’$ even mean equations (1) and (2)? None of terms in expectation depend on $M’$. Please clarify.

3. The MDPs $M$ and $M'$ appear to be totally different except sharing their action spaces. What does $V^M_{\pi^+}(s)$ mean?

4. Table 1 is not very clear to me. What is the difference between static and bounded rewards? Are they mutually exclusive and is one better than the other?

5. Can you provide a comparison between the reward perturbation made by your algorithm vs prior methods?

---

> ### Author Response · Authors · 2024-11-22
> **Response to Reviewer pHDQ**
>
> ## Summary
>
> We would like to thank the reviewer for their feedback since it has helped us greatly improve the quality of our work. Throughout this rebuttal we will refer to the updated and much improved manuscript which we have already uploaded. The reviewer’s main concerns were with our threat model, novelty relative to related works, and notational rigor. We have gone through each of the reviewer’s comments, implementing relevant changes and have provided in-depth responses. In summary, in this rebuttal we have:
> * Implemented all of the reviewer’s requested improvements, greatly increasing the quality of our paper.
> * Argued against the reviewer’s claims of an unrealistic threat model - citing the multitude of prior works with the same assumptions, noting the infeasibility of a “direct policy replacement” attack, and exploring other possible attack scenarios
> * Refuted the reviewer’s claims about lacking novelty by their cited work [1] as the attack scenarios, methods, and objectives of this paper are completely orthogonal to ours.
> * Performed additional experiments on the Frogger environment along with ablations over $\beta$ on the CAGE environment.
>
> Upon reading our rebuttal we kindly ask the reviewer to consider increasing their score. We otherwise invite the reviewer to bring up any additional questions or concerns they may have about our work.
>
> ## Addressing Weaknesses
>
> **1**. Thank you for pointing out these mistakes. We have gone through and addressed each in the updated manuscript. Please refer to our questions response below for more details.
>
> **2**. We would first like to note that this level of adversarial access is required by every single paper in the backdoor attack literature in Deep RL [TrojDRL, BadRL, SleeperNets, BackdooRL, etc.], thus our threat model is strongly supported by the existing literature. In spite of this, we agree that it is important to further explain the possible threat vectors compatible with our attack. We have adjusted our threat model section to include the possibility of malicious training software, malicious cloud training, or attacks against offline RL.
>
> We now challenge the argument that an adversary can “directly modify the policy itself.” The goal of a backdoor attack is to exploit the agent at test time. This requires the poisoned agent to perform well enough to be deployed, meaning the adversary must maintain "attack stealth" (Section 3.1). The adversary cannot simply replace the agent’s policy without training an entirely new policy themselves. They must have full access to the victim’s MDP as well as the necessary compute resources to train the agent (which is hard and expensive to obtain). Accessing the MDP may be easy if it is simulated, but would be extremely difficult, costly, or infeasible in the case of real-world training environments, such as self-driving or robotic applications.
>
> Instead, a more practical approach is a universal attack like Q-Incept, which doesn’t require domain-specific engineering or direct policy manipulation. As long as the adversary understands the agent’s observation space and can create a trigger, they can successfully and stealthily execute the attack. Our theoretical guarantees and experimental results support this, making Q-Incept a viable option for stealthy backdoor attacks, regardless of the adversary’s access level.
>
> **3**. While [1] is certainly an important piece of context, we strongly disagree that diminishes the novelty of our work. First, [1] studies attacks at *testing time* against fixed policy $\pi$ that is **not being trained**. In [1] the attacker forces the agent to take some adversarial action $a^\dagger_t$ at test time according to their adversarial reward function $g(s,a,r)$. With Q-Incept, no direct manipulation occurs - the adversary manipulates the agent’s *experience data*, not the simulator or agent itself, to make them “think” they took the target action when they truly took the optimal action. This is the opposite of what [1] does. Furthermore, [1] only shows that the adversary’s strategy *can be* optimized using RL based techniques, while our work shows that adversarial inception **is optimal** without further, online tuning of the strategy.
>
> Our work also makes significant theoretical contributions. Unlike prior attacks, which rely on large reward perturbations (e.g., [SleeperNets]), we develop the first backdoor attack framework for DRL that guarantees success without such perturbations, supported by detailed proofs in the appendix.
>
> We’ve cited [1] in the related work section, but believe the distinction between our approaches should be clear.

---

> ### Author Response · Authors · 2024-11-22
> **Response to Reviewer pHQD - Continued**
>
> **4**. We thank the reviewer for highlighting their issues with our experimental results section and have taken multiple steps to resolve the issue in the updated manuscript. In particular we have: added new experiments on the Atari Frogger environment, performed ablations over the poisoning rate $\beta$ on CAGE, properly defined and explained the metrics we used for evaluation (ASR and BRR), and improved our discussion to highlight notable observations.
>
> **5**. We agree that it is important to discuss defenses in our work and have thus expanded our related work and conclusion/discussion sections accordingly. We do not directly consider any defense techniques in this paper as we are proposing a new class of backdoor poisoning attack against RL which has not been studied before. As such, no properly designed defenses against this attack exist yet. Certified defenses against poisoning attacks result in large drops in episodic return and are just starting to be studied in RL [2], and other detectors rely on large discrepancies between the agent’s predicted value in poisoned states over benign states [3]. Under adversarial inception this large discrepancy no longer exists as the agent’s value in poisoned states is directly correlated to the value of optimal actions, making the difference minimal.
>
>
> The next, natural question is “which defense do we study and evade?” We could try and minimize detection by test time, information-theoretic defenders as in [4], but this would require an entire paper's worth of analysis to properly address as in [5].
>
> In short, we believe our contributions in developing an entire new class of theoretically optimal and empirically verified backdoor poisoning attacks are sufficient for publication. We are excited to see how future works expand and develop our versatile adversarial inception framework to evade different defense approaches, and are greatly interested to see what future defense techniques our paper may motivate.
>
> ## Addressing Questions
>
> **1**. We have added a better explanation for our inclusion of $\mathbb{S}$ in the updated manuscript. Additionally, M and M’ share action spaces so the second question will not raise any issues. Every action in M is also in M’ and vice versa.
>
> **2**. Originally $\pi^+$ was defined in terms of M’ in the text. We updated the equation so the relationship between $\pi^+$ and M’ is more clear. In short, the adversary aims to find an adversarial MDP M’ which optimizes these two objectives in expectation. In our theoretical results we prove that an adversarial MDP constructed according to Section 4.2 will always solve these objectives for any benign MDP.
>
> **3**. M and M’ are actually very similar in spite of their perceived differences. See that M’ is defined directly in terms of M. In fact, M’ is exactly the same as M on all time steps excluding those poisoned, and these poisoned time steps only occur with probability $\beta$, which should always be small. Intuitively, imagine watching the agent play Q*bert - under M’ (not Q-Incept, but the MDP M’ from section 4.2) the only difference you will notice is the trigger occasionally appearing on the screen for a single frame, and the agent being more likely to take optimal actions when this happens. Under the hood the agent will also receive a slightly higher reward when this happens. Otherwise the environment interactions and everything are identical.
>
> **4**. Due to space constraints we moved the table to appendix section A.1 and clarified your question in the caption.
>
> **5**. Certainly! We have included these results in appendix Section A.3.3 along with additional analysis. In short, Q-Incept’s reward perturbations are magnitudes smaller than those seen in the unbounded versions of SleeperNets and TrojDRL. When compared to bounded SleeperNets and TrojDRL the reward poisoning magnitudes are similar. Overall the reward perturbation of Q-Incept is miniscule compared to the total returns received by each agent in the respective environments (e.g. 400 expected return in Q*bert compared to a reward poisoning magnitude of <1). This means adversarial inception is the main factor in Q-Incept’s success.
>
> ## Conclusion
>
> We again thank the reviewer for their feedback. We have answered all the reviewers' questions, resolved their issues with the paper, and have refuted their claims of lacking novelty and an unrealistic threat model. Therefore we again kindly ask the reviewer to consider increasing their score.
>
> [2] https://openreview.net/forum?id=X2x2DuGIbx
>
> [3] Chen et. al. “BIRD: Generalizable Backdoor Detection and Removal for Deep Reinforcement Learning”
>
> [4] Nasvytis et al., “Rethinking Out-of-Distribution Detection for Reinforcement Learning: Advancing Methods for Evaluation and Detection"
>
> [5] Franzmeyer et al., “Illusory Attacks: Information-Theoretic Detectability Matters in Adversarial Attacks”

---

> ### Comment · Reviewer_pHDQ · 2024-11-26
>
> Thank you for your replies! I appreciate the efforts the authors have put in improving the paper but I still think that the presentation in the paper is quite complicated, and the objectives are still not well formulated. Specifically, the optimization problem over $M'$ in eqn 1, 2 does not make much sense to me - instead the optimization variable should be clearly stated.
>
> On a side note, the paper Kiourti et. al. was one of the initial papers in backdoor attack in RL and the goal stated there is also not well articulated - in many scenarios given a target action it might not be possible to impose that action on all states without significantly lowering the value of the backdoor policy in clean environment. So, additional assumption like disjointness of occupancy space should be made to rule our such scenarios and give a proper guarantee.
>
> I also think that the experiment section can be improved by properly articulating the main takeaways and contrasting with prior work wherever appropriate. So, at this point I would maintain my score.

---

> ### Author Response · Authors · 2024-11-27
> **Reply to Reviewer**
>
> Thank you for the follow-up feedback. The reason we are optimizing over $M'$ is because this best represents what the adversary can influence in the agent's training. They cannot optimize over $\pi^+$ directly, as they can only indirectly impact the learned policy through the adversarial MDP they induce, $M'$. In short - we want the expected policy $\pi^+ \sim \mathcal{L}(M')$ given a stochastic learning algorithm $\mathcal{L}$ and adversarial MDP $M'$ to have the attack success and attack stealth properties we describe. Through our proofs we show that the optimal policy in $M'$ solves both of our objectives, therefore the learning algorithm $\mathcal{L}$ must solve our objectives in order to maximize its own objective of expected episodic return. Thus, our constructed $M'$ optimizes our objectives. Also note that this objective formulation is shared with that of SleeperNets which was just accepted to NeurIPS 2024.
>
> > "in many scenarios given a target action it might not be possible to impose that action on all states without significantly lowering the value of the backdoor policy in clean environment"
>
> In our paper we prove that this is **not true** under our construction of adversarial inception. Please see Lemmas 1 and 2 as well as Theorem 2. This is further verified by our empirical results.
>
> > "So, additional assumption like disjointness of occupancy space should be made to rule our such scenarios and give a proper guarantee."
>
> You are correct, these are the exact assumptions we make in the paper, as was stated in the beginning of section 4.3. To make these assumptions more clear we have moved them into their own subsection titled "Assumptions".
>
> > "I also think that the experiment section can be improved by properly articulating the main takeaways and contrasting with prior work wherever appropriate."
>
> We have completely rewritten the experimental results section to improve our writing based upon a similar comment you had in your original review. Does the updated version sufficiently discuss key takeaways and compare against the prior work in your opinion?

---

> ### Comment · Reviewer_pHDQ · 2024-11-27
>
> I would suggest the authors to work on these to improve the quality of the paper.
> 1. Please clearly state what the adversary can control during training time and test time. Mention the interaction protocol to be more clear.
> 2. Clearly state the optimization variables. To me it looks like they are $\pi^\dagger$ and $\delta$. On a side note, just because a prior paper accepted at NeurIPS has poorly formulated the problem statement does not mean that every other paper should follow them as a defense.
> 	- please properly formulate your objective 1 in terms of the optimization variables. it would be helpful to refer to [RL theory book](https://rltheorybook.github.io/) or [Bharti et. al](https://proceedings.neurips.cc/paper_files/paper/2022/file/5e67e6a814526079ad8505bf6d926fb6-Paper-Conference.pdf) for proper mathematical notations and equations.
>
> 3. The motivation of the problem is weak. Using a bounded reward is motivated by the fact that it is easy to detect large rewards by the defender. This is not interesting if you allow the adversary to change the state to another state that is outside the support of original MDP. Detecting this is also trivial and reduces the value of just detecting reward perturbations. Secondly, I believe there still can be interlligent ways to perturb reward by not too much to achieve backdoor attack than naive reward clipping and that need more discussion.

---

> > ### Author Response · Authors · 2024-11-27
> > **Response to Reviewer pHDQ**
> >
> > **1.** The interaction protocol is already defined in section 3.3. In short, the adversary can manipulate the agent's training trajectories $H = [(s,a,r)_t]_t^\mu$ of size $\mu$ after they have been generated during training. We do not define the test time interaction in the paper since it is domain and application specific, but the adversary needs some way of inducing the agent to observe the trigger at test time. For instance, if the trigger is a specific QR code and the agent is a real-world robot with video input, the adversary can induce the adversarial behavior by placing the QR code physically somewhere in the environment.
> >
> > The problem of how and when to activate the trigger at test time is worthy of its own study. The goal of this paper was to study *if* the adversary can implement a poisoning strategy which solves our objectives of attack stealth and success under bounded reward constraints. This formulation is agnostic to and compatible with any of the attacker's test time objectives.
> >
> > **2.** We understand your confusion and would like to restate that the adversary cannot directly control the agent's policy $\pi^+$, they can only control the data the agent's policy optimizes with respect to. Your points are valid, however, so we have updated our language in section 3.1 to better reflect the adversary's objectives and have restated the optimization problem in terms of $\pi^+$. As an aside, we want to note that all of our results are agnostic to the construction of $\delta$, it can be an arbitrary function from $S$ to $\mathbb{S}$, so long as its image is disjoint from $S$. This means the adversary can implement whichever trigger function makes the most sense in their specific application domain.
> >
> > **3.** Detecting the trigger is actually a very difficult problem. If it were easy then all the literature on poisoning attacks against RL would be invalid. Remember there are some minor disconnections between the theory of back door attacks and their practical implementations. When altering data points in the aforementioned $H$ we aren't truly changing any states in the environment, we are just altering the agent's *observations* about the environment's state. Returning to our previous example, how can you say the QR code in the agent's camera view wasn't supposed to be there? Answering this question is not trivial and is an active area of research. The problem becomes exponentially more difficult when considering attacks against offline RL where the data is collected externally.
> >
> > > "there still can be intelligent ways to perturb reward by not too much to achieve backdoor attack"
> >
> > We agree with this, one such intelligent way is to perform an adversarial inception attack :) ! When starting this project our initial explorations were into attacks as you describe, where the adversary perturbs only rewards while remaining bounded. Through our exploration we found counter-examples to prove that this is *not possible* for arbitrary MDPs. Recall the cartpole task where all rewards are always 1. Under this MDP the adversary provably *cannot* achieve attack success since they cannot induce any reward manipulation without being detected. On the other hand, under adversarial inception, the target action $a^+$ *is* optimal, solving our objectives! It just isn't uniquely optimal in this case.
> >
> > There are other counter examples you can find for MDPs with less restrictive reward functions. For instance, see the counter proofs in the appendix of SleeperNets (section 9.2). These counter examples present MDPs with rewards bounded in $[0,c]$ for which any poisoning strategy bounded within $[-c, c]$ cannot achieve attack success. Note that here the adversary is actually stronger than our reward poisoning bounds allow, and yet they still cannot guarantee attack success.
> >
> > Both of these counter proofs are valid for any $\beta < 1$ meaning the adversary cannot achieve attack success no matter how large their poisoning rate is.
> >
> > We again appreciate all of your questions and comments as they have helped us improve the writing of our paper greatly. If you have any additional questions or concerns please let us know.

---

### Author Response · Authors · 2024-11-26
**Note to all Reviewers**

Thank you again for taking the time to read and review our paper. We have uploaded a version of the paper with **changes highlighted in blue text** to make the differences easier to spot. Note that the majority of the experimental result section was also changed in terms of writing and results (as requested by reviewer pHDQ), but we chose to just highlight a few key changes like the definitions of ASR and BRR.

---

### Meta-Review · Area_Chair_Y4KC · 2024-12-21

**Metareview:**

This paper studies the well-known problem of backdoor attack against DRL agents with the additional constraint that the reward perturbation during training time must be small. The novelty of the paper does not meet the bar of an averaged ICLR paper. Artificial problems like backdoor attack has long been criticized for a lack real-world motivations. Experimental evaluation is also quite weak and only performed on simple Mujoco tasks, which further aggravate the lack of practical relevance for the proposed work.

Therefore, I recommend rejection.

**Additional Comments On Reviewer Discussion:**

NA

---

### Decision · Program_Chairs · 2025-01-22

Reject